# Nesprin-2 coordinates opposing microtubule motors during nuclear migration in neurons

Chuying Zhou[1,2], You Kure Wu[1,2], Fumiyoshi Ishidate[2], Takahiro K. Fujiwara[2], and Mineko Kengaku[1,2]

**Nuclear migration is critical for the proper positioning of neurons in the developing brain. It is known that bidirectional microtubule motors are required for nuclear transport, yet the mechanism of the coordination of opposing motors is still under debate. Using mouse cerebellar granule cells, we demonstrate that Nesprin-2 serves as a nucleus-motor adaptor, coordinating the interplay of kinesin-1 and dynein. Nesprin-2 recruits dynein–dynactin–BicD2 independently of the nearby kinesin-binding LEWD motif. Both motor binding sites are required to rescue nuclear migration defects caused by the loss of function of Nesprin-2. In an intracellular cargo transport assay, the Nesprin-2 fragment encompassing the motor binding sites generates persistent movements toward both microtubule minus and plus ends. Nesprin-2 drives bidirectional cargo movements over a prolonged period along perinuclear microtubules, which advance during the migration of neurons. We propose that Nesprin-2 keeps the nucleus mobile by coordinating opposing motors, enabling continuous nuclear transport along advancing microtubules in migrating cells.**

## Introduction

Long-distance organelle transport is driven by microtubules and their motors. The cargo organelles bind to microtubule motors and move toward the microtubule plus-end by kinesin and toward the minus-end by cytoplasmic dynein 1 (dynein hereafter) to reach its destination within the cell. As the opposing motors inevitably engage in a tug-of-war competition, the force balance of those motors has to be precisely controlled to deliver the cargo organelle rapidly to the target site (Hancock, 2014). Recent extensive studies have elucidated that cargo-specific adaptor proteins support the targeted transport of various intracellular organelles not only through selective tethering of the motor proteins but also through functional activation or inactivation of the motors (Maday et al., 2014; Cason and Holzbaur, 2022). For instance, the trafficking kinesin protein 1/2 (TRAK1/2) concurrently binds to both dynein and kinesin-1 and coordinates bidirectional movements for efficient delivery of mitochondria (Van Spronsen et al., 2013; Fenton et al., 2021; Canty et al., 2023). In a scenario of axonal vesicle transport, Fidgetin-like 1 (Fignl1) binds to dynein and kinesin-3 and mediates a tug-of-war competition of opposing motors to achieve speed control (Atkins et al., 2019). In contrast, other adaptors activate only single motor at a time while forming a multimotor complex: c-Jun N-terminal kinase-interacting protein 1 (JIP1) inactivates kinesin to initiate dynein-mediated retrograde transport in the axon (Fu and Holzbaur, 2013; Fu et al., 2014); Huntingtin (Htt)

switches between kinesin- or dynein-driven transport by phosphorylation (Colin et al., 2008). Many studies have elucidated the diverse functions of various adaptor proteins, yet the complete understanding of the mechanisms of bidirectional organelle transport is still distant.

The nucleus, the largest organelle in the cell, is also subject to microtubule-dependent transport. Protein complexes embedded in the nuclear membrane, including the linker of the nucleoskeleton and cytoskeleton (LINC) complex and the nucleopore complex, serve as cargo adaptors that link microtubule motors to the nucleus. Klarsicht/ANC-1/Syne homology (KASH) domain proteins are the components of the LINC complex which span the outer nuclear membrane and interact with actin and microtubule motors. The vertebrate KASH proteins Nesprins are ubiquitously expressed in a wide range of cell types, including neurons, neural progenitor cells, muscle cells, and tumor cells, and are known to control nuclear position and shape, nuclear migration, chromatin organization, and mechanotransduction (Mellad et al., 2011; Gundersen and Worman, 2013).

During mammalian brain development, postmitotic neurons undergo active migration from their neurogenic sites to designated locations for the foundation of functional neural circuits. A typical migrating neuron has an elongated bipolar shape formed by thin leading and trailing processes, and the nucleus in

.................................................................................................................................................................

[1]Graduate School of Biostudies, Kyoto University, Kyoto, Japan; [2]Institute for Integrated Cell-Material Science (WPI-iCeMS), Kyoto University, Kyoto, Japan.

Correspondence to Mineko Kengaku: kengaku@icems.kyoto-u.ac.jp

Y.K. Wu's current affiliation is Institute of Neuroscience, Technical University of Munich, Munich, Germany.

the cell soma undergoes microtubule-dependent transport into the leading process. The conventional view is that dynein serves as the predominant motor driving unidirectional nuclear transport along microtubules, which are mostly oriented with their minus ends toward the leading process (Tsai et al., 2007). Accumulating evidence supports that Nesprin-1 and Nesprin-2 serve as nuclear cargo adaptors anchoring dynein and kinesin-1 to the nucleus in migrating neurons. Nesprin-1/2 directly interacts with kinesin-1 (Kif5) via the LEWD motif in the nuclear membrane proximal region of their long cytoplasmic stretches (Wilson and Holzbaur, 2015). Nesprin-1/2 also interacts with dynein by unknown mechanisms possibly involving dynein-associated protein Bicaudal D2 (BicD2) (Zhang et al., 2009; Gonçalves et al., 2020; Tsai et al., 2020). In mouse cerebellar granule cells (CGCs), while dynein functions as a dominant motor, kinesin-1 generates nuclear rotation, assisting the nucleus in maneuvering through the narrow space (Wu et al., 2018). On the other hand, kinesin-1 has been reported to retard forward nuclear translocation by a tug-of-war competition with dynein during radial migration of rat cortical neurons (Gonçalves et al., 2020). Thus, it is still unclear how Nesprin-1/2 achieves directional nuclear transport by coordinating the binding and/or activity of the opposing motors.

Here, we used mouse CGCs to investigate how dynein and kinesin-1 move the nucleus via Nesprin-2. We first confirmed that directional nuclear migration in CGCs requires the involvement of both dynein and kinesin-1, as well as Nesprin-2. We identified that dynein binding to Nesprin-2 requires the putative CC1-box and spindly motifs in the membrane-proximal region, independently of kinesin-1 binding to the nearby LEWD motif. By utilizing an intracellular cargo-trafficking assay, we demonstrated that Nesprin-2 drives active bidirectional transport along microtubule tracks, which persists for a longer period than single-motor-driven transport. Moreover, the mutant molecule defective in binding to either motor not only attenuates the motor-dependent transport but also significantly impairs the activity of the opposing motor, suggesting cross-functional activation by opposing motors. Although Nesprin-2 only shows a slight bias to the microtubule minus ends, it assists forward nuclear translocation by coupling the nucleus with the forward-moving perinuclear microtubules. Our results indicate that kinesin-1 functions synergistically with dynein–dynactin–BicD2 (DDB) via Nesprin-2 to facilitate forward nuclear movements in migrating neurons.

## Results

### Both dynein and kinesin-1 are required for nuclear movements
Nuclear migration of newborn CGCs can be observed in an organotypic culture of isolated cerebellar lobes (Fig. 1 A; Umeshima and Kengaku, 2013). Observation at 30-s intervals revealed that the nucleus moved forward with frequent backward steps, suggesting a possible tug-of-war competition by opposing motors (Fig. 1, B and E; and Video 1). We asked whether dynein and kinesin-1 compete against each other by introducing dominant-negative constructs of each motor using in vivo electroporation. Consistent with our previous observation in a reaggregate

culture, dynein inhibition by overexpressing the N-terminal truncate of Lis1 (Lis1N) drastically decreased nuclear movements (Fig. 1, C, E, and F; and Video 1). If dynein functions as a predominant motor and gradually transports the nucleus in a tug-of-war competition with kinesin-1, kinesin-1 inhibition should accelerate forward migration of the nucleus as was previously observed in radial migration of neocortical neurons (Gonçalves et al., 2020). However, overexpression of the tail domain of the kinesin-1 heavy chain (Kif5B-tail) failed to accelerate nuclear movement but rather strongly inhibited it in CGCs (Fig. 1, D–F and Video 1).

The contribution of the microtubule motors to CGC migration was also confirmed in a reaggregate culture (Fig. 1 G; Umeshima et al., 2007). At 1–2 days in culture, each CGC extended a long leading process out of the cell aggregate and delivered the nucleus forward with frequent backward steps (Fig. 1, H and I; and Video 1). We confirmed our previous observation of decreased nuclear movements upon the inhibition of dynein by overexpression of either Lis1N or the coiled-coil 1 of dynactin p150 (p150-CC1) or upon the inhibition of kinesin-1 by overexpression of Kif5B-tail or the tetratricopeptide repeat domain of kinesin light chain 1 (KLC1-TPR) (Fig. 1, I and J; and Video 1; Wu et al., 2018). These results suggest that kinesin-1 plays a role in promoting nuclear transport synergistically with dynein rather than simply as a competitive motor.

### Nesprin-2 recruits dynein–dynactin–BicD2 complex and kinesin-1 via separate but spatially close motifs
To understand how dynein and kinesin-1 cooperatively transport the nucleus, we focused on Nesprin-2, a highly abundant Nesprin in CGCs that has been reported to serve as a linker between the nucleus and both dynein and kinesin-1 motors in migrating postmitotic neurons (Zhang et al., 2009). The full-length Nesprin-2G contains an N-terminal actin-binding Calponin homolog (CH) domain and a C-terminal nuclear-localizing KASH domain, flanking 56 spectrin repeats (SRs) (Fig. 2 A). The kinesin-1-binding site has been identified as the LEWD (Leu-Glu-Trp-Asp) motif near the C-terminus, but dynein-binding sites remained unclear. By the mixture-of-isoforms (MISO) analysis (Katz et al., 2010) of RNA sequencing data from CGCs, the full-length transcripts of Nesprin-2 occupied around half of the total transcripts (Fig. S1 A). We then searched for potential dynein–dynactin binding sites in the full-length Nesprin-2G. Several candidate Spindly motifs (xLF/AxE where x denotes any amino acid) and CC1-box motifs (AAxxG) were found along Nesprin-2 through sequence alignment, which have been identified in many dynein–dynactin activating adaptors (Fig. 2 A and Fig. S1 B; Reck-Peterson et al., 2018; Olenick and Holzbaur, 2019; Cason et al., 2021). To determine the binding sites for the dynein–dynactin complex, we designed a series of Nesprin-2 truncated fragments and used them for co-immunoprecipitation in HEK293 cells (Fig. 2 A). As shown in Fig. 2, B and C, the most membrane-proximal fragment (SR48-56), which contains three potential Spindly motifs, one potential CC1-box motif, and the LEWD motif, was sufficient for binding to dynein heavy chain (DHC), the dynactin subunit p150[Glued], and kinesin heavy chain (KHC). SR48-56 also recruited the dynein activator BicD2, which

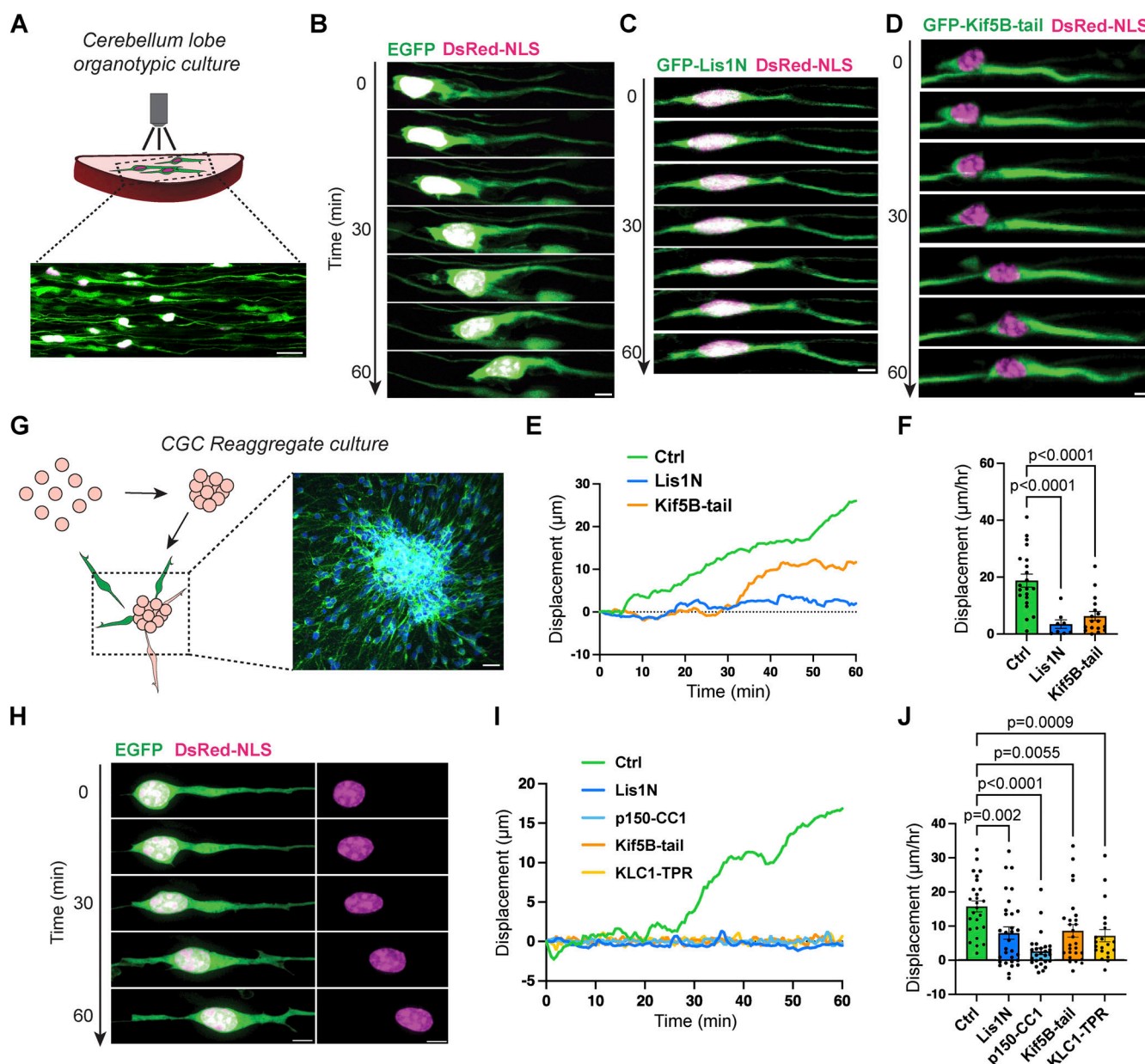

Figure 1. **Both dynein and kinesin-1 are required for nuclear movement in migrating CGCs. (A)** Schematic of live imaging set-up of cerebellar lobe culture. Plasmids were electroporated to monitor nuclear movements during neuronal migration. **(B–D)** Time-lapse sequence of a migrating CGC transfected with DsRed-labeled nuclear-localization-signal (DsRed-NLS; magenta) together with GFP (green; B), GFP-Lis1N (C), or GFP-Kif5B-tail (D). **(E)** Trajectories of nuclear centroid movements of CGC transfected with GFP (Ctrl; green), GFP-Lis1N (blue), or GFP-Kif5B-tail (orange). **(F)** Quantitative analyses of CGC nuclear net displacement. $n$ = 21 (Ctrl), 8 (Lis1N), and 17 (Kif5B-tail) neurons from three independent experiments per group. Unpaired $t$ test with Welch's correction. **(G)** Schematic illustration of CGC reaggregate culture set-up. **(H and I)** Representative time-lapse sequence of a CGC (H) and trajectories of nuclear centroid movements (I) from reaggregate cultures. **(J)** Quantitative analyses of nuclear net displacement. $n$ = 26 (Ctrl), 31 (Lis1N), 27 (p150-CC1), 27 (Kif5B-tail), and 21 (KLC-TPR) neurons from three to four independent experiments per group. Unpaired $t$ test with Welch's correction was used to compare with Ctrl. Bars show mean ± SEM. Scale bars, 20 μm in A and G; 5 μm in B–D and H. Related to Video 1.

has been reported to mediate Nesprin-2 and dynein–dynactin interactions (Gonçalves et al., 2020). Other fragments showed only weak binding to both motors and motor-associated proteins.

We next prepared the construct containing the SR48-56 and the KASH domain (SR48-KASH) and a series of its truncates and compared their binding affinities to the dynein–dynactin-BicD2 (DDB) complex and kinesin-1 by co-immunoprecipitation

(Fig. 2 D). SR49-KASH, which deleted the SR48 domain containing one putative Spindly motif from SR48-KASH, and SR52-KASH, which deleted another putative Spindly motif, exhibited progressively reduced binding to both the DDB complex and kinesin-1 (Fig. 2, D and E). Previous studies have reported that the SR52-KASH, containing a putative Spindly motif, a putative CC1 box, and a LEWD motif, is competent for binding to both the dynein–dynactin complex and kinesin-1, and for driving

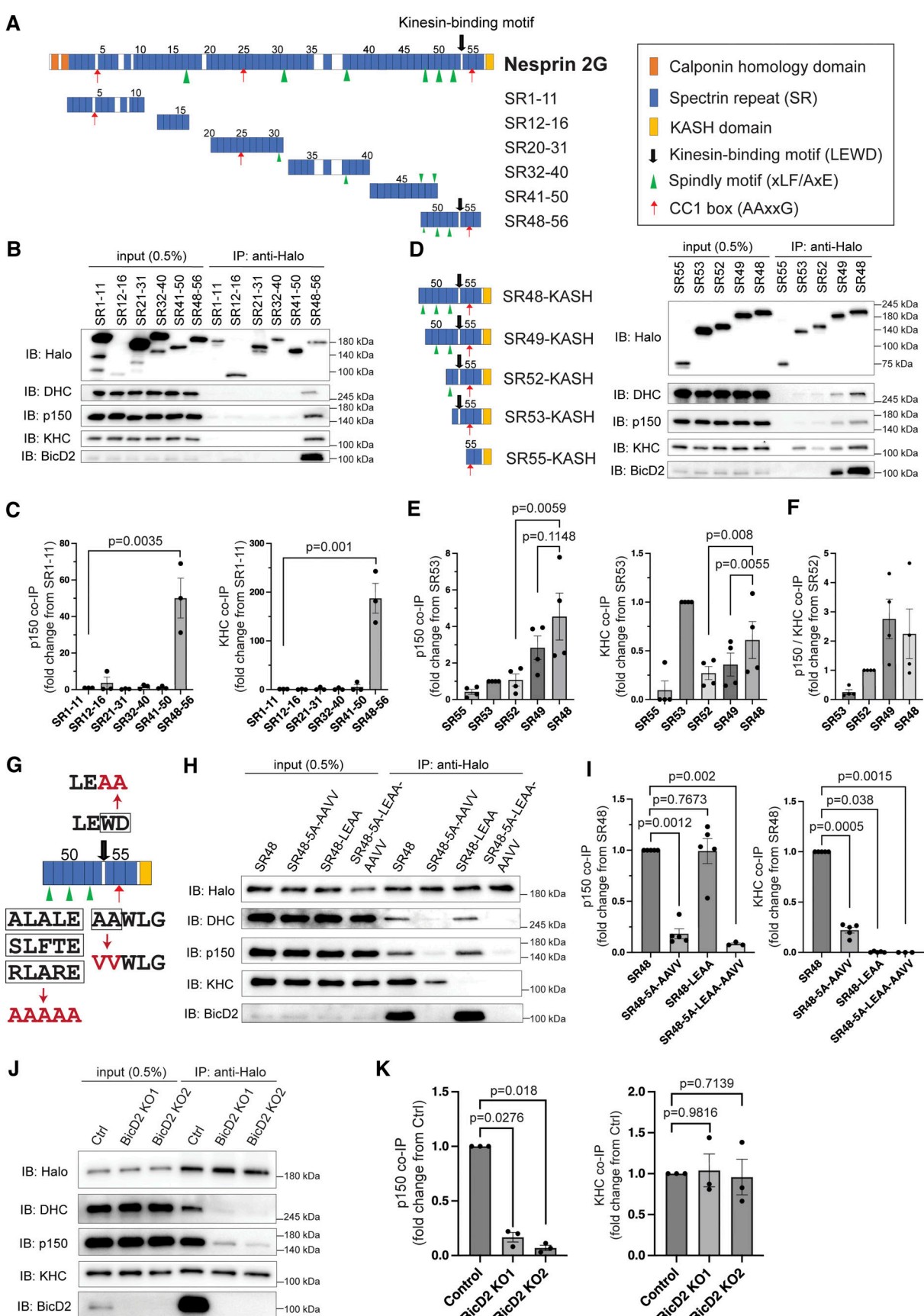

Figure 2. **Nesprin-2 recruits dynein–dynactin–BicD2 complex and kinesin-1 via separate but spatially close motifs. (A)** Schematic of Nesprin-2 Giant (Nesprin-2G) and the truncated Nesprin-2 SR fragments highlighting the kinesin-binding motif and the putative dynein-dynactin-binding motifs. **(B and C)**

Immunoblotting (B) and quantification (C) of co-immunoprecipitation (co-IP) of HEK293 cell lysate transfected with indicated Halo-tagged Nesprin-2 truncates. $n = 3$ independent experiments. Ratio paired $t$ test. **(D)** Schematic of Nesprin-2 C-terminal truncates and co-IP results of HEK293 cells transfected with indicated Halo-tagged Nesprin-2 truncates. **(E and F)** Quantification of co-IP of endogenous p150 and KHC (E) and the p150/KHC ratio (F). $n = 4$ independent experiments. Ratio paired $t$ test. **(G)** Schematic indication of the mutation sites. In SR48-5A-AAVV-KASH, all three Spindly motifs and one CC1-box motif were mutated. In SR48-LEAA-KASH, the LEWD motif was mutated. **(H and I)** Immunoblotting (H) and quantification (I) of co-IP with Halo-tagged wild-type and mutant SR48-KASH in HEK293 cell lysate. $n = 5$ independent experiments. Ratio paired $t$ test. **(J and K)** Immunoblotting (J) and quantification (K) of co-IP with Halo-tagged SR48-KASH in control (Ctrl) or BicD2 knockout (KO) HEK293 cell lysates. BicD2 KO1 and KO2 are two independently isolated KO cell lines. $n = 3$ independent experiments. Ratio paired $t$ test was used to compare with Ctrl. *IB*, immunoblotting; *IP*: immunoprecipitation. Bars show mean ± SEM. Source data are available for this figure: SourceData F2.

microtubule-based nuclear movements (Zhu et al., 2017). However, the affinity of SR52-KASH to the dynein–dynactin complex and kinesin-1 was several folds weaker than SR48-KASH. The SR53-KASH containing a putative CC1 box and a LEWD motif, but no potential Spindly motif, strongly recruited kinesin-1, but only weakly bound to dynein-related molecules (Fig. 2, D and E). The shortest construct SR55-KASH containing only a putative CC1 box motif barely showed any binding to either motor. When comparing the relative ratio of dynactin p150[Glued] over kinesin-1 binding, SR49-KASH and SR48-KASH showed a stronger bias to dynein–dynactin than the previously reported SR52-KASH (Fig. 2 F).

An extended coiled-coil of ~30 nm flanked by an N-terminal CC1-box and a C-terminal Spindly motif is a shared characteristic with many known dynein activators that might help with docking onto the 37-nm Arp1 filament of the dynactin complex (Urnavicius et al., 2015, 2018). Protein structure prediction by ColabFold indicated that SR48-KASH contains a rod-shape region formed by SRs from SR48 to SR53, which is absent in SR52-KASH (Fig. S1 C and Video 2) (Mirdita et al., 2022). The predicted rod-shaped region from SR48 to SR53 is ~33 nm, but the CC1-box-containing SR55 is separated from the Spindly-motif-containing region by a disordered region containing the LEWD motif (Fig. S1 C). The region scores low on a coiled-coil prediction tool and has the putative CC1-box and Spindly motifs in opposite positions compared with other known dynein activators (Fig. S1 D; Ludwiczak et al., 2019). Although not in a typical arrangement, we wonder whether these motifs might be involved in binding to the DDB complex. To examine this possibility, we introduced point mutations in the conserved amino acids on the putative Spindly (xLF/AxE to AAAAA/5A) and CC1-box (AA to VV) motifs (Fig. 2 G). The 5A-AAVV mutant exhibited little or no binding to BicD2, DHC, and p150[Glued] (Fig. 2, H and I), supporting that these motifs are critical for binding to the DDB complex. Kinesin-1 binding of the 5A-AAVV mutant was significantly lower than the wild-type SR48-KASH, suggesting that Nesprin-2-DDB binding is not competitive but rather stabilizes Nesprin-2-kinesin-1 binding. We also generated SR48-KASH with point mutations in the kinesin-1-binding LEWD motif (SR48-LEAA) and confirmed that SR48-LEAA failed to bind to kinesin-1 (Fig. 2, G–I). It has previously been reported that the LEAA mutant of SR52-KASH exhibits very weak binding to BicD2 (Gonçalves et al., 2020). In contrast, SR48-LEAA bound to the DDB complex at a comparable level to the control, indicating that SR48-KASH binds to the DDB complex independently of kinesin-1 and the LEWD motif.

It has previously been reported that BicD2 directly binds to SR52-KASH and recruits both dynein and kinesin-1 to Nesprin-2 in migrating cortical neurons (Gonçalves et al., 2020). To further investigate the mechanism of Nesprin-2-binding to bidirectional motors, we generated BicD2-deficient HEK293 cell lines using the CRISPR/Cas9 system and performed co-immunoprecipitation assays. In the absence of BicD2, the amount of DHC and p150[Glued] co-precipitated with SR48-KASH was greatly reduced, supporting that BicD2 is required for the binding of the dynein–dynactin complex to Nesprin-2 (Fig. 2, J and K). It also suggests that the atypical Spindly and CC1 box motifs on SR48-KASH are not sufficient for direct recruitment of dynein and dynactin in the absence of BicD2. In contrast., kinesin-1 was precipitated with SR48-KASH in BiCD2-deficient cells at levels similar to control. These results indicate that BicD2 is required for the binding of Nesprin-2 to the dynein–dynactin complex but not to kinesin-1.

Taken together, the DDB complex and kinesin-1 are thought to be recruited by Nesprin-2 via different motifs in a non-competitive manner, negating that Nesprin-2 mediates the selective binding of a dominant motor to facilitate unidirectional nuclear transport. It is also unlikely that kinesin-1 assists dynein motor activity by strengthening DDB complex binding to Nesprin-2.

### Nesprin-2 binding with both dynein and kinesin-1 is required to rescue migration defects in Nesprin-2 mutant cells

To further investigate Nesprin-2 function in nuclear migration in CGCs, we generated a mouse line deficient of Nesprin-2G by the improved genome-editing via oviductal nucleic acids delivery (i-GONAD) method (Ohtsuka et al., 2018; Gurumurthy et al., 2019). Since Nesprin-2 is a giant protein with a number of isoforms, we designed a CRISPR-based guide RNA targeting the common region shared by all the nuclear-localizing isoforms in the C-terminal SR54 region near the KASH domain. We selected a mouse line harboring a 16-bp deletion of the 108th exon in *Nesprin-2* gene, which led to a predominantly reduced level of Nesprin-2G (Fig. S2, A–C). Immunostaining of cerebellar tissue sections and cultured CGCs from homozygous mutant mice showed only traces of diffusive Nesprin-2 signals in the cytoplasm while their wild-type littermate showed clear Nesprin-2 signal decorating the nuclear envelope (Fig. S2, D and E).

In Nesprin-2 mutant mice, the size of the cerebellum was significantly smaller than control wildtype mice at P9 and P12 when CGC migration was active, while it became comparable after cortex formation was completed by P30 (Fig. S2 F). At P9 and P12, we found significant thickening of the inner external

granule layer (EGL) populated with p27-Kip1-positive post-mitotic CGCs in Nesprin-2 mutant mice, suggesting delayed CGC migration from the EGL to the IGL (Fig. 3, A–C). We further recorded CGC migration in in vitro reaggregate culture and confirmed that Nesprin-2 mutant cells showed slower nuclear migration (mean displacement 7.46 ± 9.68 µm per hour) than wild-type cells (mean displacement 12.91 ± 10.24 µm per hour) (Fig. 3, D and E; and Video 3). Nesprin-2 mutant cells also showed weakened nucleus-centrosome coupling and decreased nuclear rotation, indicating disconnection between the nucleus and microtubule motors (Fig. S2, G and H). These results support that Nesprin-2 is required for nuclear migration in CGCs.

We next performed rescue experiments by transfecting constructs of wild-type and mutant SR48-KASH conjugated with mNeonGreen (mNG) in a primary culture of CGCs from Nesprin-2 mutant mice. The nuclear envelope of transfected cells showed a strong mNG signal (Fig. 3 F), indicating the successful integration of SR48-KASH in the nuclear membrane. The displacement of nuclei of Nesprin-2 mutant CGCs transfected with wild-type SR48-KASH resumed to the level of wild-type cells, suggesting that SR48-KASH is sufficient for restoring adaptor activity of Nesprin-2 (Fig. 3, E–G and Video 3). This result agrees with the previous report by Gonçalves et al. (2020) that the actin-binding activity of the Nesprin-2 N-terminal CH domain is not required for rescuing neuronal migration. On the other hand, transfection of non-DDB-binding 5A-AAVV mutant or non-kinesin-binding LEAA mutant failed to rescue nuclear migration (Fig. 3, E–G and Video 3). Taken together with the results in Fig. 1, it is conceivable that the anchorage of both dynein and kinesin-1 to Nesprin-2 is essential for forward nuclear movement in CGCs.

## Nesprin-2 mediates bidirectional cargo movements along microtubules

To understand how Nesprin-2 regulates the motility of dynein and kinesin-1 motors, we utilized a drug-inducible peroxisome assay using rapamycin-induced heterodimerization of FKBP and FRB (Fig. S3 A; Kapitein et al., 2010). We first confirmed that microtubules are almost uniformly arranged with their plus-end orienting toward the cell peripheral in COS7 cells (Fig. S3, B–D). COS7 cells were transfected with a fusion construct of peroxisome-targeting signal PEX3, the green fluorescent protein (GFP), and FKBP (PEX-GFP-FKBP) together with an FRB construct fused with either an active form of kinesin-1 motor Kif5B or dynein-associated protein BicD2. Consistent with previous reports, FRB fused with a truncated form of Kif5B (Kif5B-HA-FRB) quickly moved peroxisomes to the cell periphery upon treatment of rapalog, indicating the activation of plus-end-directed cargo trafficking (Fig. 4, A and B; and Video 4). On the other hand, FRB fused with the N-terminus of dynein activator BicD2 (HA-BicD2-FRB) moved peroxisomes to the cell center where microtubule minus ends were located (Fig. 4, A and B; and Video 4). Peroxisomes terminated at the cell periphery by active Kif5B or the cell center by BicD2N, respectively, and their movements declined within 30 min after rapalog addition (Fig. 4 C). We next constructed the SR48-SR56 of Nesprin-2 including the DDB and kinesin-1 binding domains (SR48-KASH

with deletion of the KASH domain) that are fused with FRB (HA-SR48ΔKASH-FRB) and co-transfected it with PEX-GFP-FKBP. We anticipated that by recruiting dynein and kinesin-1 motors, the SR48ΔKASH would generate bidirectional movements on microtubules. SR48ΔKASH indeed induced active and prolonged bidirectional peroxisome movements both toward and away from the cell center after rapalog addition (Fig. 4, A and B; and Video 4). SR48ΔKASH exhibited persistent directional switches, resulting in no obvious bias toward either the cell periphery or center. SR48ΔKASH also demonstrated longer-lasting active transport than Kif5B and BicD2 (Fig. 4 C).

We next examined SR48 mutants in the peroxisome transport assay. The SR48-LEAA mutant, which binds to the DDB complex but not to kinesin-1, exhibited very limited transport activities to both ends of microtubules (Fig. S3, E and F; and Video 5), supporting that kinesin-1-binding enhances rather than competes with dynein-mediated transport. In contrast, the SR48-5A-AAVV mutant, which is unable to bind to the dynein complex, moved peroxisomes toward the cell periphery (Fig. S3, E and F; and Video 5), but more slowly than Kif5B. These results suggest that Nesprin-2-mediated transport toward the microtubule minus-end requires both dynein and kinesin-1 binding, while the plus-end-directed transport requires only kinesin-1.

To further characterize motor kinetics mediated by Nesprin-2, we traced single peroxisomes moving along microtubule tracks by dual-color time-lapse imaging of peroxisomes and microtubules at sub-second time resolution. MRC5 cells were used because of the sparse distribution of microtubules and flat morphology. Microtubules in MRC5 cells were mostly radiating from the cell center to the periphery (Fig. S3, B–D). Peroxisomes exhibited short processive runs interspersed by pausing or detaching from microtubules. Most Kif5B- and BicD2-driven cargoes showed unidirectional transport to the microtubule plus- and minus-ends, respectively (Fig. 4, D and E; and Video 6). In contrast, Nesp2-SR48ΔKASH-FRB generated a mixed population of bidirectional and unidirectional transport along microtubule tracks (Fig. 4 F and Video 6).

We analyzed the velocities by identifying the active transport segments on kymographs, where peroxisomes moved with constant speed for over 1 µm distance. The velocities and run length of those processive segments are quantified by the semi-automated KymoAnalyzer tool (Neumann et al., 2017). Previous studies on single-molecule imaging and intracellular cargo transport have reported that kinesin-1 moves at velocities of 0.5–0.8 µm/s while the DDB complex exhibited a broader distribution of velocities ranging from 0.9–2 µm/s (Thorn et al., 2000; Seitz and Surrey, 2006; Nath et al., 2007; Ross et al., 2008; Lu et al., 2009; McKenney et al., 2014; Schlager et al., 2014; Elshenawy et al., 2019; Htet et al., 2020; Serra-Marques et al., 2020; Gicking et al., 2022). The velocities measured from our experiments are within the range of the previously reported speed (Fig. 5 A). Furthermore, FRB fused with dynein heavy chain (FRB-HA-DHC) moved to the minus-end at lower velocities and run length than HA-BicD2N-FRB, supporting that BicD2 enhances the stability of dynein–dynactin complex and promotes its motor activity (Fig. 5, A and B; and Fig. S3 G). Nesp2-SR48ΔKASH-FRB generated bidirectional movements at around

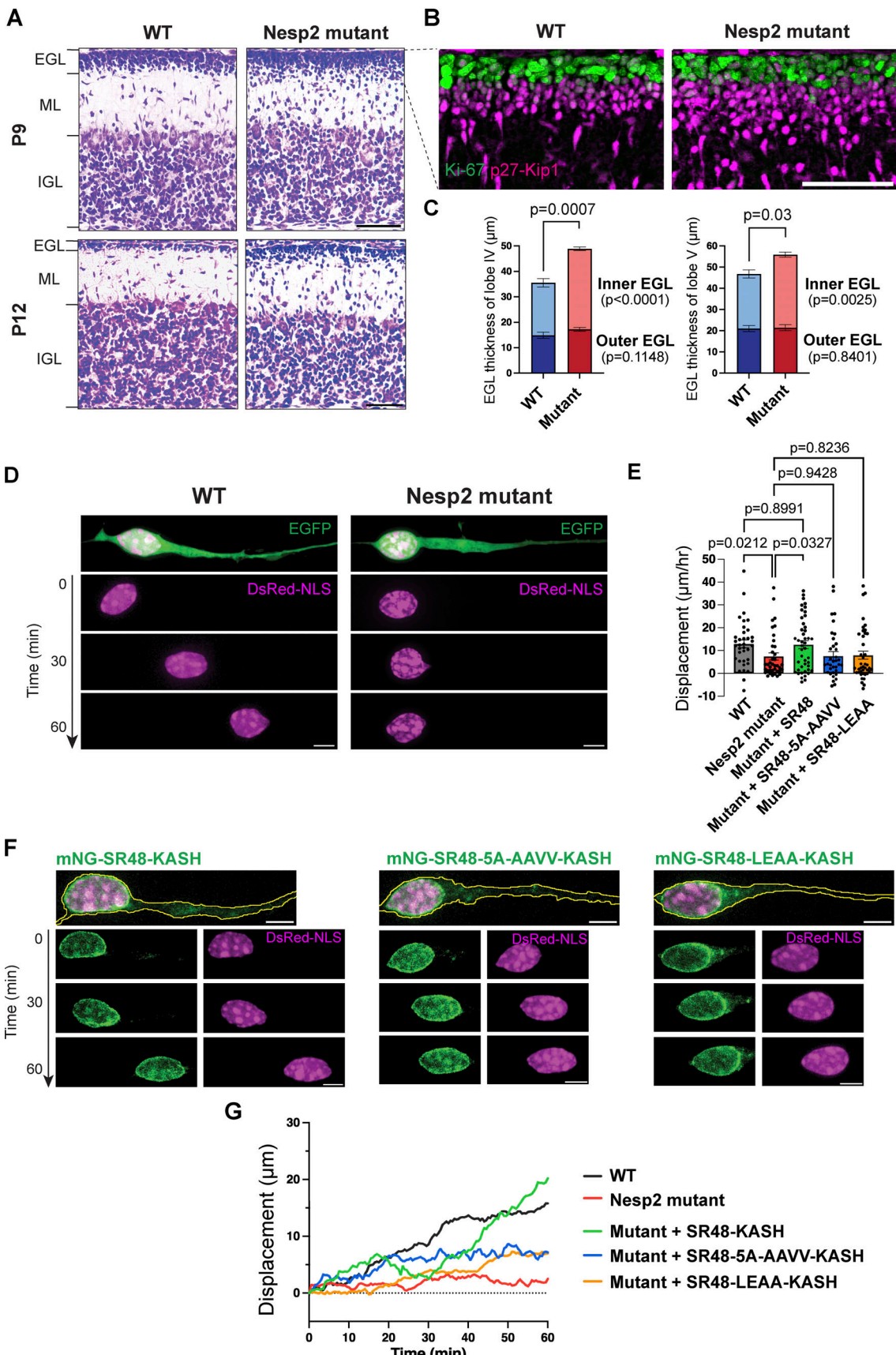

Figure 3. **Nesprin-2 needs to bind to both dynein and kinesin-1 to drive nuclear migration in CGCs. (A)** Nissl staining of the cerebellum lobe IV from WT and mutant mice at P9 and P12. Post-mitotic CGCs migrate from the external granule layer (EGL), through the molecular layer (ML), eventually reaching the

inner granule layer (IGL). **(B)** Immunofluorescence of Ki-67 (green) and p27-Kip1 (magenta) in the EGL from WT and Nesprin-2 mutant mice. **(C)** Quantitative analysis of the thickness of the outer and inner EGL of the cerebellar lobe IV (left) and lobe V (right). *n* = 5 (WT) and 7 (mutant) littermate mice from three independent experiments. Unpaired *t* test was used to compare WT and mutant. **(D)** Representative time-lapse sequences of CGCs in reaggregate culture from WT and mutant mice. Cells were transfected with EGFP and DsRed-NLS. **(E)** Quantitative analysis of nuclear net displacement. *n* = 36 (WT), 38 (Nesp2 mutant), 45 (Mutant + SR48), 36 (Mutant + SR48-5A-AAVV), and 41 (Mutant + SR48-LEAA) cells from four to five independent experiments per group. Unpaired *t* test. **(F)** Representative time-lapse sequences of CGCs from Nesprin-2 mutant mice, which were transfected with DsRed-NLS together with mNG-SR48-KASH, mNG-SR48-5A-AAVV-KASH, or SR48-LEAA-KASH. **(G)** Trajectories of nuclear centroid in cells that are shown in D and F. Bars show mean ± SEM. Scale bars, 50 μm in A and B; 5 μm in D and F. Related to Video 3.

1 μm/s in both directions (Fig. 5 A), suggesting that SR48ΔKASH recruited both DDB and kinesin-1. Interestingly, association with SR48ΔKASH enhanced the plus-end-directed cargo transport speed by kinesin-1, while the minus-end-directed transport by SR48ΔKASH was comparable with BicD2. The averaged run length of SR48-mediated transport to minus- and plus-ends were indistinguishable, and slightly lower than BicD2- or Kif5B-driven transport, respectively (Fig. 5, B and C). The frequency of minus-end-directed transport was slightly higher than plus-end-directed transport (60 ± 10%) (Fig. 5 G), suggesting a biased movement toward the minus end.

The cargo dynamics mediated by SR48-LEAA and SR48-5A-AAVV mutants were also analyzed. Although the mutants exhibited much less active peroxisome movements overall, we recorded occasional processive transport when imaging at sub-second time interval. The non-kinesin-bound LEAA mutant showed a strong bias to minus-end transport, while the non-dynein-bound AAVV mutant exhibited a strong bias to plus-end transport (Fig. 5 G; and Fig. S3, H and I). Both mutants showed slower velocities than wild-type Nesp2-SR48ΔKASH in either minus-end- or plus-end-directed transport, respectively (Fig. 5 D), indicating less efficient transport in the absence of the opposing motor. The mutants did not change the run length of cargo transport (Fig. 5, E and F). We concluded that Nesp2-SR48ΔKASH enhances transport velocities by kinesin-1 and dynein in the presence of both motors, but single-motor-bound Nesprin-2 mutants are incapable of fully activating the working motor.

Since the cargo transport by Nesp2-SR48ΔKASH was only slightly biased toward the microtubule minus-end direction, this alone cannot account for the forward-biased movement of the nucleus in migrating neurons. It has been demonstrated that multiple dyneins exert a larger collective force than multiple kinesins (Rai et al., 2013). As the large nucleus might recruit a large number of Nesprins, which might mediate the cooperative function of associated motors, we wonder whether increasing the cargo size would change the directional preference of Nesprin-2-mediated transport. We tested this possibility by increasing the size of peroxisomes with mVenus-SKL (Ser-Lys-Leu) overexpression (Efremov et al., 2014). We performed a cargo transport assay using COS7 cells transfected with mVenus-SKL in which the peroxisome size was increased by 1.6 folds (Fig. S4, A–C and Video 7). However, we did not see an increased bias of Nesprin-2-mediated transport to the cell center. Overexpression of the PEX-mCherry-GM130^C-ter (C-terminus of Golgi matrix protein 130), which artificially ties peroxisomes in a row and increased its size by 3.8 fold, also showed little effect on the bias of trafficking direction (Fig. S4, A–C and Video 7; Fukumitsu

et al., 2015). These results did not support the cooperative interplay of motors bound to multiple Nesprin molecules.

## Nuclear translocation is accompanied by forward movements of microtubule tracks

To further understand the cargo motilities generated by Nesprin-2 molecules in migrating neurons, we performed the peroxisome assay in cultured CGCs. Kif5B caused rapid movement of cargoes distally in the bipolar processes, where microtubule plus ends were supposed to orient (Fig. 6 A and Video 8). On the other hand, activation of BicD2-mediated transport quickly confined cargoes to the microtubule organizing center (MTOC), which was validated by immunostaining of γ-Tubulin (Fig. 6 B, Fig. S5 A, and Video 8). Consistent with the results using the cell lines, Nesp2-SR48ΔKASH drove active bidirectional peroxisome movements in the cell soma and the leading process of migrating CGCs (Fig. 6 C and Video 8). We confirmed that peroxisomes moved along microtubules by dual-color imaging with intracellular microtubules labeled by GFP-tagged doublecortin (DCX-GFP) (Fig. 6, D–F and Video 9). In contrast to the transport by BicD2 or Kif5B, which rapidly converged at the MTOC or the neurite tips in 30 min, SR48ΔKASH maintained peroxisome motility for >50 min of observation (Fig. 6 G).

While the peroxisomes moved bidirectionally in the cell soma, they gradually advanced as the nucleus migrated forward (Video 8). These observations prompted us to hypothesize that the cargo organelles are not simply transported along stationary microtubule tracks by motors, but they are dynamically anchored onto the microtubules that are actively moving forward. To test this, we observed the movements of microtubules by using a photo-convertible mEos2 protein conjugated with DCX. DCX labeled thick acetylated microtubule bundles in the leading process and separated from the thinner bundles in the trailing process (Fig. S5 B). The movement of a small segment of microtubules demarcated by photoconversion of mEos2-DCX was observed along with the nucleus labeled with SiR-DNA dye (Fig. 7 A and Video 10). In wild-type cells, forward movements of the nucleus were often found to be coupled with forward sliding of microtubules (Fig. 7 B). In contrast, in Nesprin-2 mutant CGCs, nuclear movements were hampered, but microtubules still moved forward, leaving the nucleus behind (Fig. 7 C). More severe disconnection between the nucleus and microtubules was observed in CGCs expressing the dominant negative KASH construct (Fig. 7 D), which was reported to block both Nesprin1 and 2 (Stewart-Hutchinson et al., 2008). Photoconverted MT segments moved forward at comparable speed among all groups, but the nuclear-microtubule (NM) distance gradually increased

**Figure 4. Nesprin-2 drives bidirectional cargo transport along microtubules. (A)** Inducible peroxisome trafficking assay in COS7 cells transfected with PEX-GFP-FKBP together with KIF5B-HA-FRB (top), HA-BicD2-FRB (middle), or HA-SR48ΔKASH-FRB (bottom). Images at 0, 30, and 60 min after rapalog treatment are shown. GFP signals are shown in black and cell contours are outlined in blue. Right: Representative trajectories of GFP signals are shown with

lines by different colors, with time denoted by color gradient and ending points denoted by filled circles. **(B)** Distribution of GFP fluorescence along the distance from cell center to periphery at different time points. Kif5B- or BicD2-expressing cells show rapid PEX-GFP displacement towards the cell edge or cell center, respectively. Nesprin-2-SR48-expressing cells show persistent fluctuation of GFP distribution, indicating continuous bidirectional cargo transport. **(C)** Quantification of the number of peroxisomes that moved 3 μm or more in respective time windows. The values are normalized to the number of moving peroxisomes in the first time window (0–5 min). $n$ = 9 (Kif5B), 16 (BicD2), and 16 (SR48) cells from three independent experiments per group. **(D–F)** Representative time-lapse sequences (left) and kymographs (right) of rapalog-induced peroxisome transport by Kif5B (D), BicD2 (E), and Nesp2-SR48 (F). Images were taken at 0.1 s interval in MRC5 cells with microtubules oriented from minus (left) to plus (right) ends. Kymographs from different experiments are shown by different colors and the leftmost kymograph in each panel corresponds to the peroxisome shown in the time-lapse images. Data points and error bars show mean ± SEM. Scale bars, 10 μm in A; 2 μm in D–F. Related to Videos 4 and 6.

in Nesprin-2 mutant and KASH-overexpressing cells (Fig. 7, E and F). Together, our data suggest that rather than simply dragging the nucleus toward microtubule minus-ends on fixed tracks, Nesprin-2 keeps the nucleus in a bidirectionally movable state while coupling it with forward-moving microtubules (Fig. 7 G).

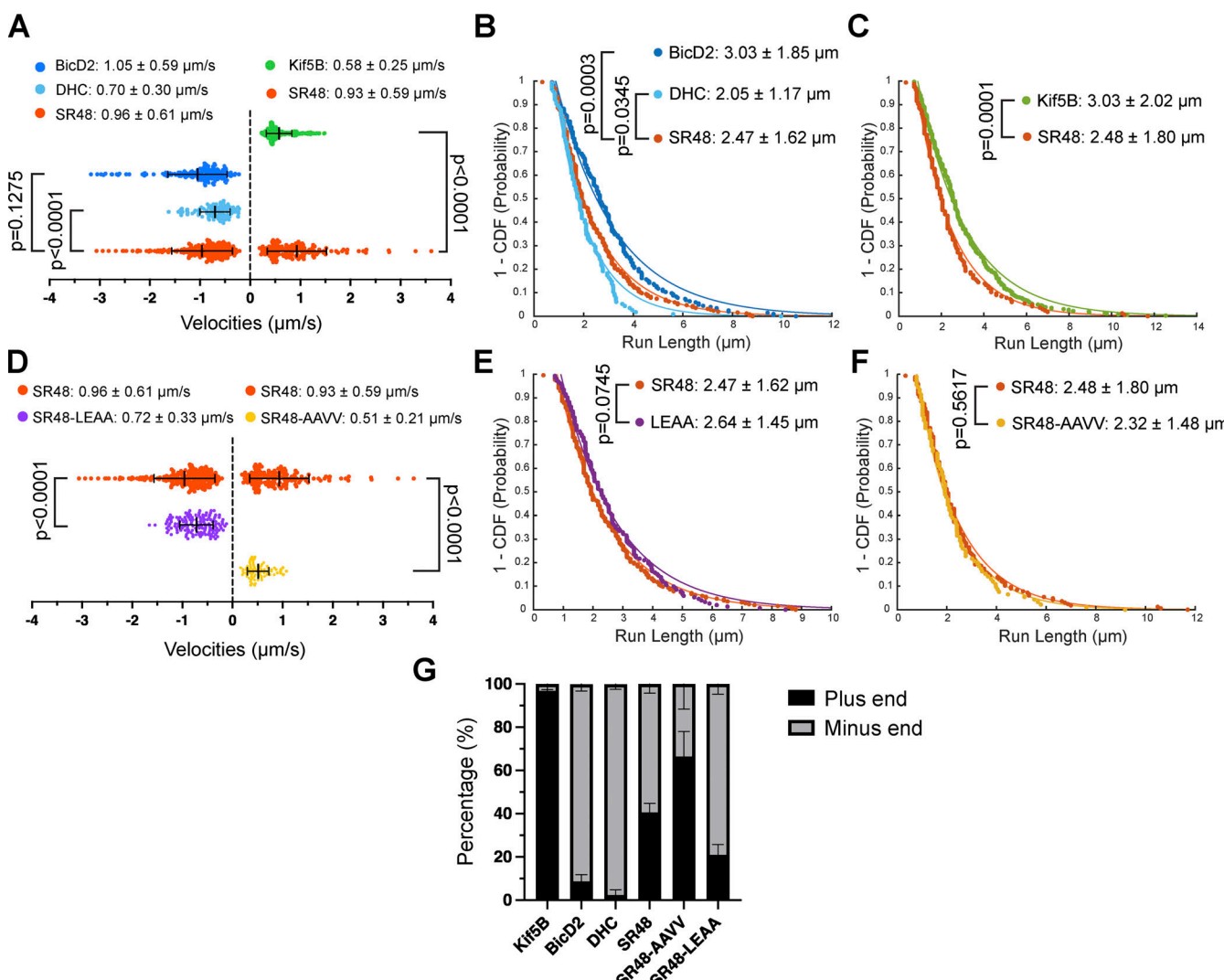

Figure 5. **Kinetics of cargo transport driven by Nesprin-2 SR48. (A)** Velocity distribution of peroxisomes transported by Kif5B, BicD2, DHC, and Nesp2-SR48. Velocities from processive transport segments with a distance >1 μm are included. Positive velocities indicate movement toward MT plus ends and negative velocities indicate movement toward MT minus ends. $n$ = 343 (Kif5B), 188 (BicD2), 103 (DHC), 147 (SR48 to MT plus end), and 226 (SR48 to MT minus end) trajectory segments in >25 cells from three to five independent experiments per group. **(B and C)** Inverse cumulative distribution functions (CDF) of run length toward MT minus end (B) and plus end (C). Lines show the fitted single exponential decay curves and data points show the actual values. **(D)** Velocity distribution of peroxisomes transported by Nesp2-SR48-LEAA and Nesp2-SR48-5A-AAVV compared to Nesp2-SR48. $n$ = 147 (SR48 to MT plus end), 226 (SR48 to MT minus end), 143 (SR48LEAA), and 82 (SR48-AAVV) trajectory segments in >25 cells from three to five independent experiments per group. **(E and F)** Inverse cumulative distribution functions (CDF) of run length toward MT minus end (E) and plus end (F). **(G)** Percentages of plus- and minus-end-directed runs. Unpaired $t$ test with Welch's correction is used in A and D. Mann Whitney test is used in B, C, E, and F. Bars show mean ± SEM. Numerical values are mean ± SD.

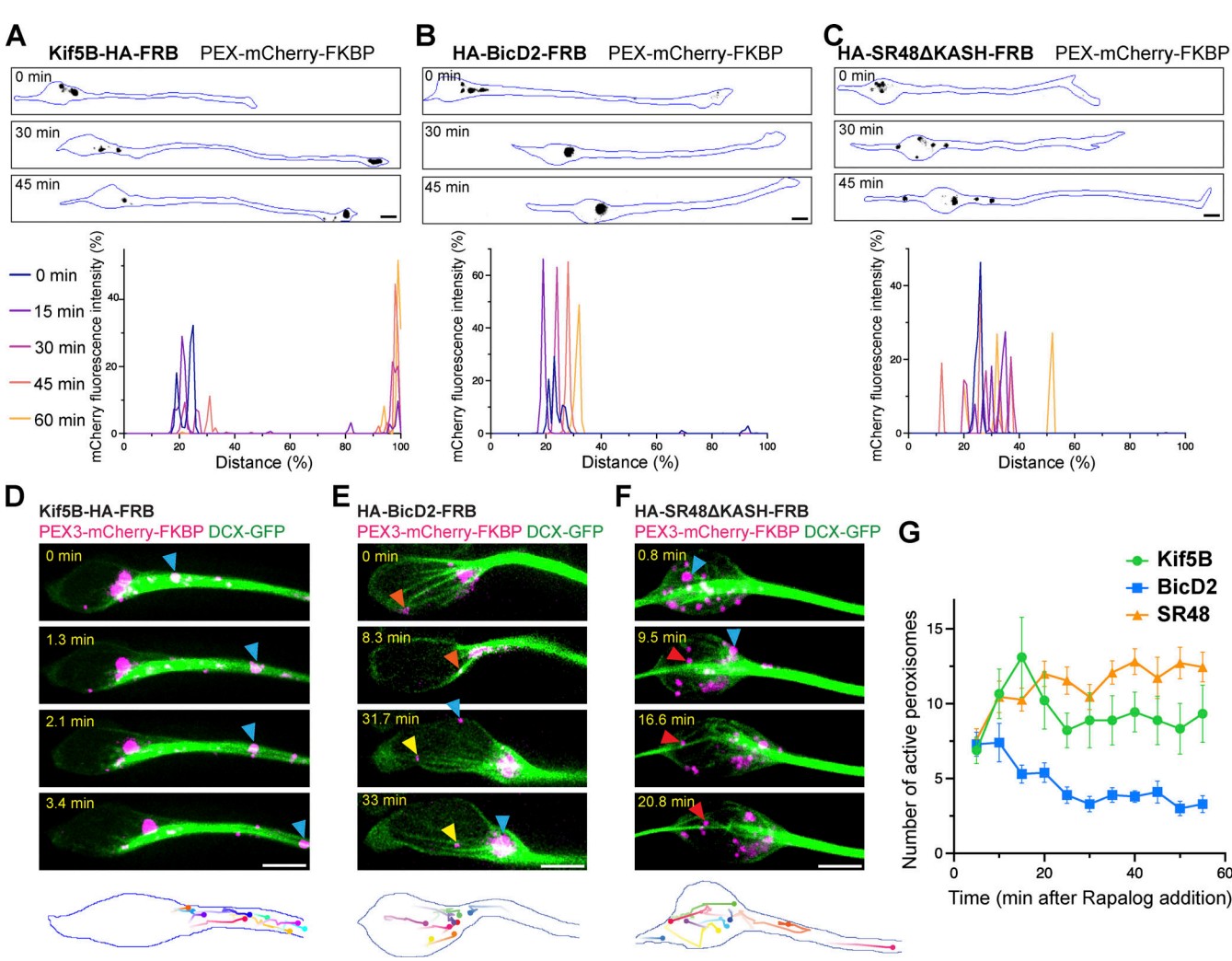

**Figure 6. Nesprin-2 drives persistent bidirectional cargo transport in CGCs. (A–C)** Inducible peroxisome trafficking assay in cultured CGCs transfected with PEX-mCherry-FKBP together with KIF5B-HA-FRB (left), HA-BicD2-FRB (middle), or HA-SR48ΔKASH-FRB (right). Images at 0, 30, and 60 min after rapalog treatment are shown. mCherry signals are shown in black and cell contours are outlined in blue. Lower graphs show the distribution of mCherry fluorescence along the distance from the trailing process to the leading process at different time points. Kif5B- and BicD2-expressing cells showed rapid PEX-mCherry displacement toward the tip of the leading process and the microtubule-organizing center, respectively. Nesprin-2-SR48-expressing cells showed persistent fluctuation of PEX-mCherry distribution, indicating continuous bidirectional cargo transport. **(D–F)** Representative dual-color time-lapse sequences showing peroxisomes (magenta) moving along DCX-GFP-labeled MT filaments (green) in CGCs. Colored arrowheads trace the movement of individual peroxisomes. Bottom, trajectories of individual peroxisomes. **(G)** Quantification of the number of peroxisomes that moved 3 μm or more in respective time windows. n = 9 (Kif5B), 11 (BicD2), and 11 (SR48) cells from three independent experiments per group. Data points and error bars show mean ± SEM. Scale bars, 5 μm. Related to Videos 8 and 9.

## Discussion

### Nesprin-2 recruits both dynein and kinesin-1 during nuclear transport in migrating neurons

In this study, we demonstrate the function of Nesprin-2 as a nuclear adaptor protein that recruits both dynein and kinesin-1 during nuclear migration in newly born CGCs. Through the Nesprin-2 adaptor, kinesin-1 and dynein do not hinder each other like tug-of-war, instead, they rather cooperatively move the nucleus forward along with advancing microtubules. Together with our previous study by Wu et al. (2018), we showed that nuclear migration was significantly impeded upon inhibition of either dynein or kinesin-1 in vivo and in vitro, suggesting positive involvement of the opposing motor kinesin-1 in forward nuclear transport. While our results suggest a different conclusion from Gonçalves et al. (2020), who reported accelerated nuclear translocation in kinesin-inhibited rat cortical neurons, differences in neuronal types and developmental stages might be part of the reasons for the discrepancy. Many neurodevelopmental diseases caused by defective neuronal migration have been mapped to mutations on dynein or dynein accessory proteins like LIS1 (Feng and Walsh, 2001). Studies of kinesin-1 deficiency have reported brain malformations and severe impairment of mitochondrial and endo-lysosomal trafficking. However, defects in neuronal migration have been underappreciated, which might be due to the embryonic lethality of constitutive deletion of Kif5B or functional redundancy of other kinesin family members (Tanaka et al., 1998; Kanai et al., 2000; Falzone et al., 2009; Cromberg et al., 2019).

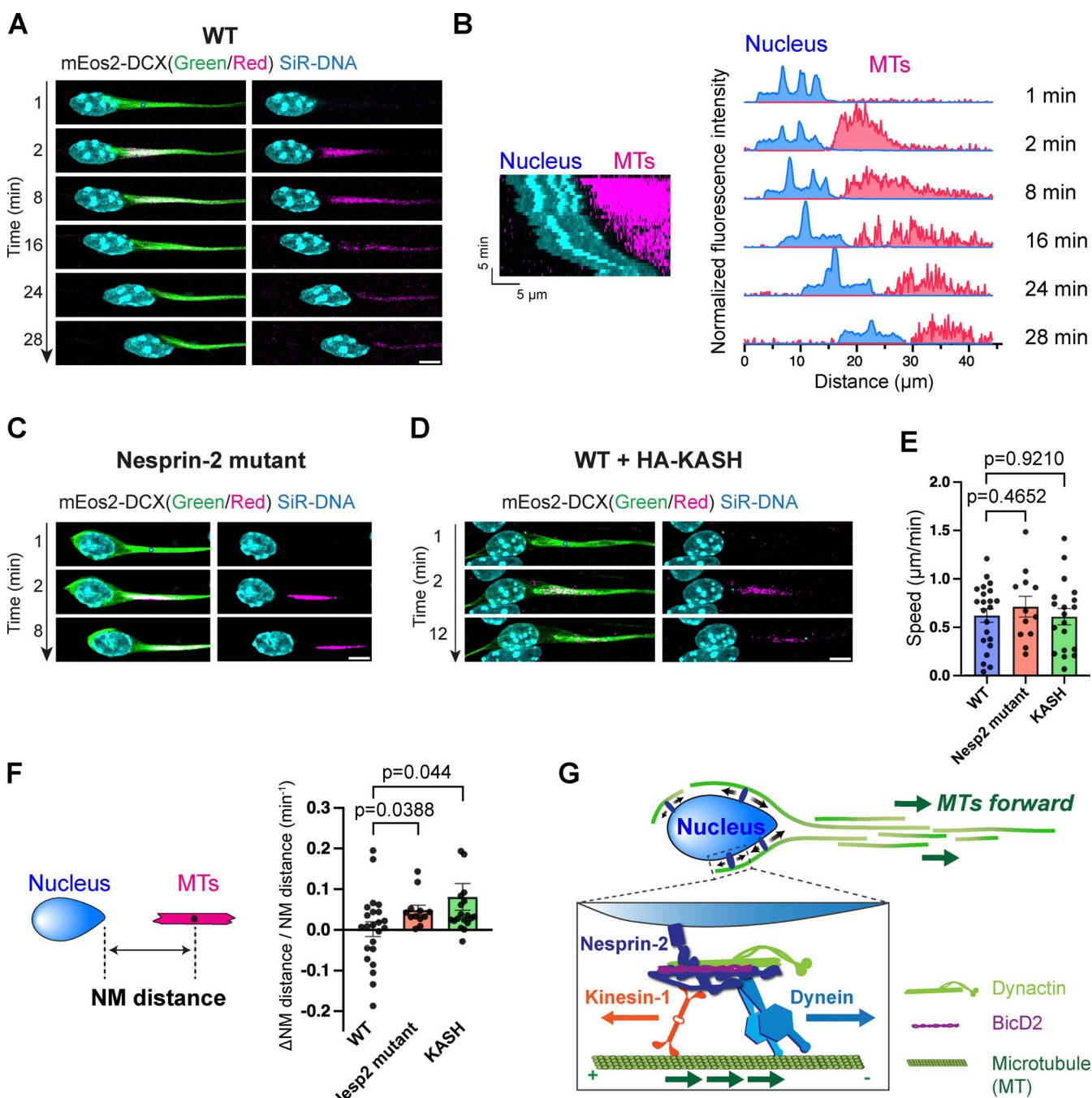

Figure 7. **Nuclear translocation is accompanied with forward movements of microtubule tracks. (A)** Representative time-lapse sequence of a cultured CGC transfected with mEos2-DCX. The nucleus was labeled with SiR-DNA. At 2 min, a strong 405-nm laser is applied at a point (denoted by a blue circle) in the proximal region of the leading process for green-to-red photoconversion of mEos2-DCX. **(B)** Kymograph (left) and intensity distribution (right) of fluorescent signals of the cell in A show synchronous forward movement of the nucleus (blue) and photoconverted MTs (red). **(C and D)** Representative time-lapse sequences of Nesprin-2 mutant CGCs (C) or WT CGCs overexpressing HA-KASH (D). Photoconversion is induced at 2 min at the points demoted by blue circles. **(E)** Quantification of MT forward moving speed. Unpaired *t* test is used to compare to WT. **(F)** Quantification of changes in Nuclear-MT (NM) distance in migrating CGCs. The change of NM distance over 3–6 min is normalized to the initial NM distance. Unpaired *t* test with Welch's correction to Ctrl samples. *n* = 22 (WT), 12 (Nesp2 mutant), and 19 (WT + HA-KASH) cells from two to six independent experiments per group. **(G)** Model of nuclear translocation mechanism. Forward movements of nucleus are generated by the combination of the persistent Nesprin-2-mediated bidirectional transport and MT forward movements in CGCs. Bars show mean ± SEM. Scale bars, 5 µm. Related to Video 10.

In Nesprin-2 mutant mice, we observed abnormal cerebellar layer formation, characterized by delayed migration of post-mitotic CGCs. It is in line with earlier studies showing that Nesprin-2-deficient mice exhibited mislocalized nuclei in the retina and disrupted layer formation in the cerebral cortex (Zhang et al., 2009; Yu et al., 2011). Mutation of Nesprin-2 has also been implicated in human intellectual disability and autism diseases, which might be related to retarded neuronal migration as well (Young et al., 2021). Although we targeted the gene region shared by all the nuclear-localizing isoforms of Nesprin-2, our mutant model did not completely abolish nuclear movements, possibly due to complementary function by other Nesprin family proteins such as Nesprin-1 (which interacts with dynein), Nesprin-4 (which interacts with kinesin-1), or other nuclear envelope proteins like RAN binding protein 2 (RanBP2), which interacts with the cytoskeleton (Roux et al., 2009; Taiber et al., 2022; Yi et al., 2023). By comparing wild-type and mutant SR48-KASH in rescue experiments, we demonstrated that the recruitment of both dynein and kinesin-1 onto the nuclear envelope via Nesprin-2 is critical for normal nuclear movements.

## Nesprin-2 independently and simultaneously recruits the dynein complex and kinesin-1

Sequence alignment and site-specific mutation clarified close but distinct motifs for kinesin-1 and DDB binding in the membrane-proximal domain of Nesprin-2. The Spindly and CC1-box motifs are found in many dynein activators, with a CC1-box flanking at the N-terminus and a Spindly motif flanking the C-terminus of an extended coiled-coil domain that mediates interaction with dynein light intermediate chain and the pointed end complex of dynactin, respectively (Gama et al., 2017; Canty and Yildiz, 2020). In Nesprin-2, the positions of the putative motifs are reversed, and their arrangement is complex in the predicted conformation. We also demonstrate that BicD2 is required for the interaction between Nesprin-2 and dynein–dynactin complex, while the presumptive Spindly and CC1-box motifs on SR48-KASH are not sufficient for direct recruitment of dynein or dynactin, which suggest differential roles compared with the previously characterized Spindly and CC1-box motifs. Although the detailed mechanism remains elusive, our results indicate that the interaction between Nesprin-2 and DDB complex is dependent on these Spindly- and CC1-box-like sequence motifs, possibly by mediating BicD2 recruitment or maintaining certain protein conformation for DDB binding. On the other hand, unlike the previous studies using SR52-KASH, kinesin-1 binding of SR48-KASH was unaffected by depletion of BicD2, suggesting that Nesprin-2 associates with kinesin-1 probably by direct binding to kinesin light chain (KLC) via the LEWD motif (Wilson and Holzbaur, 2015; Wu et al., 2018).

Notably, the predicted interface of DDB complex interaction encompasses the kinesin binding motif on Nesprin-2, similar to other bidirectional motor adaptors including JIP1 and TRAK2 (Fu and Holzbaur, 2013; Fu et al., 2014; Van Spronsen et al., 2013; Fenton et al., 2021; Canty et al., 2023). Those bidirectional motor adaptors scaffold the complex involving dynein-dynactin and kinesin, while differentially regulating the interplay of motors. JIP1 activates one motor for unidirectional transport toward either microtubule end. In contrast, TRAK2 readily switches from one motor to the opposing motor on a single microtubule. While Nesprin-2 SR48ΔKASH is sufficient to induce both plus-end and minus-end-directed cargo transport, short episodes of transport to either direction are more prevalent than directional switching in our intracellular cargo transport assay. In between those processive transports, peroxisomes pause on the microtubule or detach and jump to another microtubule. Whether the switching depends on stochastic binding or more sophisticated regulation is unknown. Unidentified accessory proteins or kinases might play a role in determining the directional switch. In fact, Klar (Drosophila KASH protein) was reported to coordinate the bidirectional transport of the lipid droplet by forming a complex with Halo and LSD2 as molecular toggles (Welte et al., 1998, 2005).

## The possible mechanisms of bidirectional cargo transport by Nesprin-2

We found that the velocity of minus-end-directed transport by Nesprin-2 is comparable with that of BicD2-driven transport, suggesting that Nesprin-2 may rely on the DDB complex for minus-end-directed cargo transport. On the other hand, Nesprin-2 enhances the velocity of plus-end-oriented transport compared to Kif5B, suggesting its function as an activating adaptor for kinesin-1 similar to Nesprin-4, TRAK1, and JIP1 (Chiba et al., 2017, 2022; Henrichs et al., 2020). Our observation is in contrast to the tug-of-war model which predicts reduced velocities for transport in either direction in the presence of both motors (Belyy et al., 2016), or the selective recruitment or selective activation model, which predicts unchanged velocities of individual motors (Hendricks et al., 2010). We assume that Nesprin-2 coordinates interdependent functions of bidirectional motors. Such interdependent coordination has been demonstrated for TRAK2, which alters landing rates in the presence of opposing motors (Fenton et al., 2021; Canty et al., 2023). Several in silico studies also support that the involvement of multiple distinct motors affects the association of the cargo with the microtubules, altering frequency and/or run length (Müller et al., 2010; Jensen et al., 2021). In contrast, Nesprin-2 bound with bidirectional motors showed comparable run lengths to the mutants bound to only a single motor, suggesting that Nesprin-2 may mediate activation or disinhibition of the active motor by the opposing motor, rather than changing association with the microtubules. The Nesprin-2-motor complex lacking either motor might be in an unfavorable conformation to release the remaining motor from autoinhibition, and thus walk at suboptimal speed. In fact, motor velocities can be changed by adaptor molecules and intra-/inter-molecular tension between closely packed motors (Yildiz et al., 2008; Elshenawy et al., 2019). Besides, mechanical activation by opposing pulling force has also been proposed as an alternative interdependent mechanism, awaiting further evidence (Ally et al., 2009; Hancock, 2014). In our intracellular cargo transport assay, the frequency of active cargo transport and cargo interaction with microtubules were hard to quantify due to the variation of peroxisome abundance and local microtubule density inside the cell. Future study is needed to determine whether the binding of the opposing motor

as a structural scaffold or force generation by the opposing motor as a mechanical activation cue is required for motility activation.

The nucleus is a large cargo and should simultaneously associate with a large number of motors. It has been reported that motor stepping and load-bearing capacity change when larger cargoes are carried, or stronger resistant forces are applied (Rai et al., 2013; Pyrpassopoulos et al., 2020). In our cargo transport assay, the directional preference of Nesprin-2-mediated transport was not affected by increasing the cargo size, negating that Nesprin-2 mediates the generation of collective force of bound motors. However, it is challenging to determine the exact number of bound or active motors and the ratio of opposing motors in our assay. Stalling forces exerted by bidirectional motors via Nesprin-2 are also unknown. Single-molecule imaging, biophysical experiments, like optical trap, and computational modeling will be helpful in providing a further understanding of how Nesprin-2 modulates motilities, force generation, or attachment/detachment rates of dynein and kinesin-1.

### How does the bidirectional adaptor Nesprin-2 mediate directional nuclear transport in neurons?

We demonstrated in both cell lines and migrating CGCs that Nesprin-2 generated prolonged bidirectional transport compared with Kif5B- or BicD2-driven peroxisome transport. The latter quickly terminated at the cell periphery or cell center in a short time, resembling an "exhausted" state. In fact, in vitro studies have demonstrated that activated DDB or Kif5 motors do not dissociate quickly but strongly accumulate at microtubule minus or plus ends (Tan et al., 2018; Chiba et al., 2022). The persistence of movements generated by alternating dynein- and kinesin-1-driven transport might be beneficial for the navigation of the large nucleus through a crowded intracellular environment. Nesprin-2 may facilitate track switching at microtubule intersections or directional reversal when it encounters intracellular obstacles. We assume that nuclear rotation, which is decreased in Nesprin-2-deficient neurons, is an important aspect of nuclear movement dynamics for smooth transport (Wu et al., 2018).

Another key question is how the net direction of transport is achieved with a bidirectional adaptor. We demonstrate that perinuclear microtubule tracks, anchored to the nucleus via Nesprin-2, move forward in migrating CGCs. It has already been demonstrated that microtubules in the leading process move forward during neuronal migration at a speed comparable with our observation (0.62 ± 0.33 versus 0.66 µm/min in Trivedi et al. [2017]). Forward movements of microtubules in neurons have been shown to be mediated by actin-microtubule crosslinking (Hutchins and Wray, 2014; Trivedi et al., 2017), or by dynein- and kinesin-driven sliding (Rao et al., 2016; Muralidharan et al., 2022; Schelski and Bradke, 2022). We observed that in migrating CGCs, the nucleus and associated microtubule bundles advance simultaneously, but this was disrupted in Nesprin-2-deficient cells.

Taken together, Nesprin-2 incorporates the motilities of dynein and kinesin-1 motors to generate persistent bidirectional cargo movements. Unlike the prevailing view that dynein acts as a predominant motor propelling nuclear transport along stationary microtubules, we propose that the nucleus continues small-step bidirectional movements along the forward-moving perinuclear microtubules. Our study provides a novel perspective on the long-unsolved paradox of motor co-dependence. In the future, characterization of Nesprin-2 transport at a single molecular level or protein structure analysis of Nesprin-2-motor complex should reveal further mechanisms of regulation.

## Materials and methods

### Animals

All animals were treated in accordance with the guidelines of the Animal Experiment Committee of Kyoto University. Wild-type ICR mice (Slc:ICR) were obtained from Japan SLC. During experiments, anesthesia was performed by ice-induced cryoanesthesia for neonatal mice, or isoflurane inhalation for adult mice to minimize pain.

### Plasmids and reagents

Constructs used in this study are listed in Table 1. SRs of mouse Nesprin-2 were defined according to protein domain information on UniProt (UniProt Consortium, 2023) and NCBI databases. Nesprin-2 fragments were cloned from a mouse cDNA library and inserted between the attL1 and attL2 sites on pENTR1A vector (Invitrogen) to be fused with different protein tags through the Gateway cloning method. Destination vectors were constructed by inserting the Gateway cassette containing the attR1 and attR2 sites into the pCAGGS vector (pCAG-Dest). Halo tag was amplified from the pHTN HaloTag CMV-neo vector (Promega) and fused with an HA tag before being inserted into the pCAG-Dest vector (pCAG-Halo-HA-Dest). mNG sequence was amplified from pmNeonGreen-C1 and inserted into the pCAG-Dest vector (pCAG-mNG-Dest). Gateway LR recombination was performed between pENTR1A-Nesp2-SRs plasmids and pCAG-Halo-HA-Dest or pCAG-mNG-Dest plasmids using Gateway LR clonase (Invitrogen) to make plasmids expressing Nesprin-2 fragments with Halo-HA or mNG tag as listed in Table 1. Site-specific mutations on SR48-5A-AAVV or SR48-LEAA plasmids were introduced using primers containing the desired mutations by In-Fusion Cloning (Takara). For the peroxisome trafficking assay, the original FKBP and FRB plasmids are gifts from Lucas Kapitein, from which we cloned the FKBP and FRB to construct the related plasmids listed in Table 1. Kif5B (1–807aa), BicD2 (1–595aa), and tubulin alpha 1b (α-tubulin) were cloned from a mouse cDNA library. SR48ΔKASH (5655–6767aa) was cloned from the pENTR-SR48KASH plasmid. pGW2-FRB-HA-DHC was made by replacing the GFP on pGW2-FRB-GFP-DHC (gift from Kapitein) with an HA tag. FRB and FKBP constructs were expressed by pGW vectors containing a CMV promoter in COS7 and MRC5 cells, while in granule cell culture, FRB and FKBP constructs were expressed by pCAG vectors containing a CAG promoter for better expression in neurons. mVenus-SKL was constructed by fusing the mVenus tag with a short sequence expressing Ser-Lys-Leu (5′-TCCAAACTC-3′) and inserting it into the pGW1 vector. For the photoconversion

Table 1.  **Plasmids**

| Plasmid | Source |
|---|---|
| pCAG-EGFP | Wu et al. (2018) |
| pCAG-DsRed-NLS | Wu et al. (2018) |
| pCAG-GFP-Lis1N | Wu et al. (2018) |
| pCAG-GFP-Kif5B-tail | Wu et al. (2018) |
| pCAG-Halo-HA-Nesp2-SR1~11/12~16/21~31/32~40/41~50/48~56 | This paper |
| pCAG-Halo-HA-Nesp2-SR55/53/52/49/48-KASH | This paper |
| pCAG-Halo-HA-Nesp2-SR48-5A-AAVV-KASH | This paper |
| pCAG-Halo-HA-Nesp2-SR48-LEAA-KASH | This paper |
| BICD2 CRISPR/Cas9 KO plasmid | Santa Cruz (sc-405525) |
| pCAG-mNG-SR48-KASH | This paper |
| pCAG-mNG-SR48-5A-AAVV-KASH | This paper |
| pCAG-mNG-SR48-LEAA-KASH | This paper |
| pCAG-GFP-EB3 | Wu et al. (2018) |
| pGW1-PEX3(1–42aa)-GFP-FKBP | This paper |
| pGW1-Kif5B(1–807aa)-HA-FRB | This paper |
| pGW1-HA-BICD2(1–595aa)-FRB | This paper |
| pGW1-SR48ΔKASH-FRB | This paper |
| pGW1-SR48-5A-AAVV-ΔKASH-FRB | This paper |
| pGW1-SR48-LEAA-ΔKASH-FRB | This paper |
| pAAV-CAG-mCherry-α-tubulin | This paper |
| pGW2-FRB-HA-DHC | This paper |
| pGW1-mVenus-SKL | This paper |
| pAAV-CAG-PEX-mCherry-GM130[C-ter] | Fukumitsu et al. (2015) |
| pCAG-PEX3-mCherry-FKBP | This paper |
| pCAG-Kif5B(1–807aa)-HA-FRB | This paper |
| pCAG-HA-BICD2(1–595aa)-FRB | This paper |
| pCAG-HA-SR48ΔKASH-FRB | This paper |
| pCAG-DCX-GFP | Wu et al. (2018) |
| pCAG-mEos2-DCX | This paper |
| pCAG-HA-KASH | Umeshima et al. (2007) |

experiment, DCX was cloned from pCAG-DCX-GFP and fused with mEos2, followed by insertion into the pCAG vector.

### Organotypic and primary culture of cerebellum granule cells

Transfection by in vivo electroporation and cerebellum lobe culture was performed as described previously with a few modifications (Umeshima et al., 2007; Umeshima and Kengaku, 2013). Briefly, DNA solution was injected into the cerebellum of P5 mice and electric pulses were applied by a pulse generator (CUY21; Nepagene). At P6, mice were dissected and the cerebellum lobes with fluorescent signal were isolated carefully in artificial cerebrospinal fluid (aCSF) bubbled with a gas mixture of 95% $O_2$ and 5% $CO_2$. The isolated cerebellar lobes (lobe IV, V, or VI) were then placed on a Millicell cell culture insert (Millipore) and covered by a thin layer of collagen gel. Lobe culture was maintained in culture medium (5.5 g/l; Basal Medium Eagle [BME] [Sigma-Aldrich], 25% Earle's Balanced Salt solution [Sigma-Aldrich], 15% heat-inactivated horse serum [Thermo Fisher Scientific], 5.6 g/l glucose, 3 mM GlutaMAX Supplement [Gibco], 1.8 g/l sodium bicarbonate, 1 mM sodium pyruvate, 1% N2 supplement [Gibco] and 50 U/ml Penicillin-Streptomycin [P-S]). Time-lapse imaging to observe granule cell migration was performed 3–18 h after the lobes were plated.

Reaggregate culture was performed as described previously (Wu et al., 2018). Reaggregate culture was maintained in Mason's medium (9.2 g/l; BME [Sigma-Aldrich], 2.1 g/l sodium bicarbonate, 4.76 g/l glucose, 10 g/l; BSA, 1% N2 supplement [Gibco], and P-S). Time-lapse imaging to observe granule cell migration was performed 18–30 h after plating.

### Live-cell imaging of CGCs

Live imaging of CGCs was performed as described previously (Umeshima et al., 2007; Wu et al., 2018) with some modifications. For lobe cultures, time-lapse imaging of 30-s intervals was performed on a laser-scanning confocal microscope (FV1000-BX61WI; Olympus) with high-sensitivity GaAsP detectors. Samples were kept in a stage-top incubator to maintain the culture temperature at 37°C with a gas flow of 85% $O_2$, 5% $CO_2$, and 10% air. A 20× (N.A. 0.5) or a 40× (N.A. 0.8) water-immersion objective was used. Images were acquired through the FV10-ASW Viewer software (Olympus). For the reaggregate culture of CGCs, time-lapse imaging of 30-s intervals was performed by a spinning-disk confocal microscope system (CV1000; Yokogawa) through a 100× (N.A. 1.3) oil-immersion objective lens. Samples were kept at 37°C with 5% $CO_2$ flow. Images were acquired through the CellVoyager CV1000 software (Yokogawa).

### Cell line culture

HEK293T cells (Institute of Physical and Chemical Research BRC Cell Bank) were maintained in DMEM (Thermo Fisher Scientific) supplemented with 10% heat-inactivated FBS and P-S. COS7 and MRC5 cells (Institute of Physical and Chemical Research BRC Cell Bank) were maintained in 1:1 of DMEM: Ham's F-10 Nutrient Mix (Thermo Fisher Scientific) supplemented with 10% heat-inactivated FBS and P-S. All cells were passaged every 3–4 days and kept on surface-treated culture dishes (Nunc dishes by Thermo Fisher Scientific or TC-treated dishes by Corning).

### Co-immunoprecipitation and Western blot

Co-immunoprecipitation was performed according to the datasheet of HaloLink Resin (Promega) with some modifications. HEK293T cells were transfected with Lipofectamine 2000 for 40–48 h before harvest. Cells were lysed on ice with PHEM buffer (50 mM PIPES, 50 mM HEPES, 1 mM EGTA, 2 mM $MgSO_4$, and Halt protease and phosphatase inhibitor cocktail [Thermo Fisher Scientific]) supplemented with 1% Digitonin (FUJIFILM Wako Pure Chemical). Cell lysate was cleared by centrifuging at 17,000 $g$ at 4°C for 15 min. The supernatant was

**Table 2. Antibodies and reagents**

| Antibody or reagent | Catalog numbers/Source | Application and dilution |
|---|---|---|
| Lipofectamine 2000 | Invitrogen | Transfection |
| Anti-Halo | G9281 (Promega) | WB 1:1,000 |
| Anti-DYNC1H1 (DHC) | 12345-1-AP (Proteintech) | WB 1:500 |
| Anti-p150[Glued] | 610473 (BD Biosciences) | WB 1:1,000 |
| Anti-UKHC F-5 (KHC) | Sc-133184 (Santa Cruz) | WB 1:1,000 |
| Anti-BICD2 | Ab117818 (Abcam) | WB 1:1,000 |
| Anti-Nesprin2 | PA5-78438 (Invitrogen) | WB 1:1,000; IHC and ICC 1:200 |
| HRP-conjugated anti-β-actin | Sc-47778 (Santa Cruz) | WB 1:1,000 |
| Anti-p27-Kip1 | 610241 (BD Biosciences) | IHC 1:400 |
| Anti-Ki-67 | 14–5698-82 (eBioscience) | IHC 1:1,000 |
| Anti-laminB1 | Ab16048 (Abcam) | ICC 1:1,000 |
| Anti-γ-tubulin | T5192 (Sigma-Aldrich) | ICC 1:250 |
| Anti- β-tubulin | T4026 (Sigma-Aldrich) | ICC 1:1,000 |
| Anti-acetylated-tubulin (6-11B-1) | T6793 (Sigma-Aldrich) | ICC 1:1,000 |
| SiR-DNA | CY-SC007 (Cytoskeleton) | Live-cell imaging 200 nM |

collected for protein concentration measurement by protein assay BCA kit. An equal amount of protein was prepared as input samples and each sample was incubated with 50 µl of HaloLink Resin at 4°C for 2–4 h. Beads were washed with PHEM buffer with 1 mg/ml BSA to reduce non-specific binding. SDS sample buffer was used for elution and the results were analyzed through SDS-PAGE and Western blot.

For Western blot, membranes were cut to stain with anti-DHC, p150, KHC, and BicD2 antibodies separately. Stripping was performed using WB Stripping Solution (Nacalai tesque) before reprobing with anti-Halo antibodies to check pull-down efficiency. The primary antibodies used are listed in Table 2. HRP-conjugated secondary antibodies (Bio-Rad Laboratories) were used, followed by a chemiluminescence reaction using the Amersham ECL Select kit (Cytiva). Blot images were taken by a gel imager (LAS-4000; FUJIFILM) with the ImageQuant LAS 4000 Control Software. To analyze co-immunoprecipitation results, the protein band intensity was measured on FIJI and normalized to the intensity of Halo-tagged protein.

### Generation of BicD2 knockout cell lines
Knockout cell lines were generated according to Ran et al. (2013) and Han et al. (2022) with a few modifications. HEK293T cells were transfected with BICD2 CRISPR/Cas9 KO plasmid (Table 1) by Lipofectamine for 2 days prior to sorting. Cells were dissociated into single cells, and FACS sorting was performed to isolate the GFP-positive cells. The sorted cells were replated onto culture dishes sparsely and allowed to expend for 10 days. Cell colonies were identified by visual inspection and 10 colonies were transferred into separate culture dishes for further expansion. SDS-PAGE and Western Blot were performed using lysate from the isolated cell clones to examine knockout efficiency, among which 5 out of 10 clones showed depletion of BicD2 protein. The two cell clones with faster growth rates were used in co-immunoprecipitation experiments.

### Generation and validation of Nesprin-2 mutant mouse line
Nesprin-2 mutant mouse line was established by i-GONAD method using ICR mice as described by Ohtsuka et al. (2018) and Gurumurthy et al. (2019), and genotyping was performed according to Bhattacharya and Van Meir (2019). CRISPR RNA targeting the 108th exon of Syne2/Nesprin-2 (crRNA sequence 5′-GTAAAGCTGCTATTACGTCAAGG-3′) and trans-activating crRNA (tracrRNA sequence 5′-CCTTGACGTAATAGCAGCTTT AC-3′) were designed using a web tool CHOPCHOP (Labun et al., 2019). RNP complexes of gRNA duplexes and Alt-R S.p. Cas9 Nuclease V3 (Integrated DNA Technologies) were injected into the oviduct of 0.7-day pregnant mice and then electric pulses were applied by electroporator (NEPA21). The electroporated mother mice gave birth to G0 founders, and genotyping was performed with forward (5′-GTTGTTGGGAATTGTTCACAGA-3′) and reverse (5′-CAGACACCAAGCTCCACATATC-3′) primers. The G0 mice carrying insertion or deletion were mated with wild-type mice to obtain G1 offspring. The PCR products from G1 mouse genotyping were inserted into the pGEM-T vector (Promega) by TA cloning (TOYOBO) for sequencing. The mouse line carrying a 16-bp deletion was selected, which generated a frameshift and a novel premature stop codon 30-bp downstream from the start position of the 108th exon. Therefore, the mutant mice either express a truncated form of Nesprin-2 without the C-terminal KASH domain or have an overall reduction of Nesprin-2 translation by nonsense-mediated decay of Nesprin-2 mRNA. For Western blot, lysate of cerebellum tissue from P9 pups was prepared in RIPA buffer and analyzed using NuPAGE 3–8% Tris-Acetate protein gel (Thermo Fisher Scientific) to visualize protein bands with large molecular weights. After validation of knockout, heterozygous mice were intercrossed to obtain homozygous wild-type and mutant mice for histology phenotyping and primary culture. Mice were used without distinction between male and female.

### Immunohistochemistry and analysis of cerebellum morphology
For brain tissue sample preparation, anesthetized mice were perfused with 4% PFA in phosphate buffer (PB). Dissected brains were postfixed for 3 h with 4% PFA in PB and dehydrated in 15% sucrose in PBS for 6–10 h, and then in 30% sucrose for overnight at 4°C. Brains were embedded in O.C.T. compound for snap freezing. Mid-sagittal sections of 10 µm thickness were prepared using Cryostat (CM1950; Leica). For immunohistochemistry, slices were treated with HistoVT One (Nacalai tesque) at 70°C for 20 min for antigen retrieval. After permeabilization and blocking, slices were incubated with primary antibodies

overnight at 4°C and then with Alexa Fluor secondary antibodies (Thermo Fisher Scientific) at 4°C for 8–10 h. The primary antibodies used are listed in Table 2. Images were taken with a laser-scanning confocal microscope (FV1000-BX61; Olympus) through 40× (N.A. 0.95) or 100× (N.A. 1.4) oil immersion objective lenses. For Nissl staining, cryosection slices were stained in 0.1% Cresyl Violet solution at 37°C for 5 min. After dehydration and clearing, slices were mounted with Entellan new (Sigma-Aldrich). Images were taken by an upright optical microscope (DM5000B; Leica) with 2.5× (N.A. 0.07), 10× (N.A. 0.3), and 20× (N.A. 0.5) objective lenses through an AdvanCam-U3X camera (Advan Vision) and the AdvanView imaging software.

### Immunocytochemistry of CGC primary cultures

For staining with Nesprin-2 or γ-tubulin, reaggregate culture samples were fixed with ice-cold methanol at −20°C for 5–10 min. Primary antibody incubation was performed at 4°C overnight. After washing, samples were incubated with Alexa Fluor secondary antibodies at room temperature for 1 h, followed by DAPI staining (10 μg/ml) and mounting with ProLong Gold Antifade Mountant (Invitrogen). For staining with acetylated-tubulin, samples were fixed in 4% PFA with 0.1% glutaraldehyde in PHEM buffer at room temperature for 20 min, followed by permeabilization with 0.2% Triton X-100 in PHEM buffer for 15 min. The following blocking and antibody incubation were performed in PHEM buffer with the same incubation temperature, time, and mounting method as the methanol-fixed samples. The primary antibodies used are listed in Table 2. Images were acquired by the same microscope system as described above for immunohistochemistry samples.

### Drug-inducible peroxisome trafficking assay

Peroxisome trafficking assay was performed as described by Kapitein et al. (2010) with some modifications. COS7 cells were plated on glass-bottom dishes (IWAKI) coated with 50 μg/ml fibronectin (Sigma-Aldrich). MRC5 cells were plated on glass-bottom dishes (IWAKI) coated with 0.2 mg/ml poly-D-lysine. Cells were transfected with indicated plasmids and Lipofectamine 1 day after plating. Cargo trafficking assay and live-cell imaging were performed 1 day after transfection for COS7 cells and CGCs, or 2 days after transfection for MRC5 cells. To induce peroxisome movements, A/C Heterodimerizer (also called rapalog AP21967; Clontech Laboratorie) was added at a final concentration of 500 nM in COS7 cells, 1 μM in MRC5 cells, or 100 nM in CGCs. For COS7 cells and CGCs, time-lapse imaging at 30-s or 4–5-s intervals was performed by a spinning-disk confocal microscope system (CV1000; Yokogawa) with a 40× oil-immersion (N.A. 1.3) or a 100× oil-immersion (N.A. 1.3) objective lenses. For MRC5 cells, dual-color time-lapse imaging was performed by a multimodal fast confocal microscope system (Andor Dragonfly 500) with a 100× oil-immersion (N.A. 1.49) objective lens. Images of both channels on a single z-plane were acquired simultaneously by an EMCCD camera (iXon Ultra) and a sCMOS camera (Zyla 4.2 Plus) at 102-ms intervals for 3 min in each cell, and multiple cells were imaged from the same dish

within 30 min after rapalog addition. All cells were imaged at 37°C with 5% $CO_2$ gas flow.

### Photoconversion time-lapse imaging of CGCs

Reaggregate cultures were incubated with 200 nM of SiR-DNA dye for at least 2 h before imaging. Triple-color time-lapse images were taken by a laser-scanning confocal microscope (LSM880; Zeiss) with the ZEN Black software (Zeiss) at 1-min intervals through a 40× oil-immersion (N.A. 1.4) objective lens. Photoconversion was induced by applying a strong 405-nm laser (laser power 45%) at a spot in the proximal region of the CGC leading process for one iteration (2–3 s). The magenta channel shown in the figure was the subtraction of the pixel values of the far-red 633-nm laser channel from the original 561-nm laser channel to minimize noise from the SiR-DNA signal.

### Image analysis and data presentation

All images were processed and analyzed using the FIJI software (Schindelin et al., 2012). For fluorescence images, maximum projections of multiple z stacks were used in all figures except for dual-color imaging of peroxisomes on microtubules at fast acquisition speed (Fig. 4, D–F and Fig. S3, G–I), and except for the fixed samples with LaminB1 and Nesprin-2 staining where a single Z stack was used to show the nuclear rim (Fig. S2 E). Gamma adjustment with value 0.5 was performed on images in Fig. 4, D–F and Fig. S3, G–I for better visualization of microtubule filaments. Prism 10 (GraphPad) and MATLAB R2023b were used for plotting trajectories and quantitative analysis.

To analyze nuclear movements, the centroid position of the nuclei in each time frame was detected by the Analyze Particles function on FIJI. The net displacement of the nuclei was calculated from the distance between the initial and final nuclear positions. The total travel distance was the summation of forward or backward nuclear movements between each time frame.

For peroxisome trafficking assay in COS7 and CGCs, peroxisome movements were tracked by a particle tracking tool from the MosaicSuite plug-in on FIJI (Sbalzarini and Koumoutsakos, 2005) with the following parameters: kernel radius, 0.1–1 μm; cutoff score, 0.01; intensity percentile, 0.1–0.5; particle link range, 1 frame; maximum step length, 7 μm; and Hungarian optimizer. A custom MATLAB program was used to plot the representative trajectories with the cell outline and to analyze peroxisome dynamics at different time points. The number of active peroxisomes at each time window was counted as the number of trajectories that travel in a direction within 45° parallel to the radius of the cell and travel for >3 μm. Peroxisomes at the edge of the cell were excluded due to non-specific movements caused by the active polymerization or depolymerization of microtubule filaments at the cell periphery. To analyze the distribution of peroxisomes in the cell, the pixel intensities of PEX-GFP-FKBP or PEX-mCherry-FKBP signal were summed up along the distance from the cell center to the cell edge using a custom MATLAB program. The center and outline of the cell were determined from the co-expression of mCherry-α-tubulin and the center was identified as the point where microtubules

converged in the perinuclear region. Directions of EB3 comets were analyzed using KymoButler (Jakob et al., 2019).

To analyze the velocities and run lengths of peroxisomes from fast acquisition images of MRC5 cells, peroxisomes that moved for over 1 µm along a single microtubule were manually identified and kymographs were generated by tracing the microtubule filaments with orientation from the cell center to the cell edge through a semi-automated FIJI plug-in KymoAnalyzer (Neumann et al., 2017). Quantification of velocities and run lengths was also performed using KymoAnalyzer by selecting the kymograph segments with a constant slope which were regarded as movements with constant speed. The representative kymographs shown in Fig. 4 were traced with a FIJI toolbox SNT (Arshadi et al., 2021). The values of run lengths were fitted by an exponential decay curve on MATLAB.

To analyze the microtubule movements in CGCs, a 3–6-min image where the mEos2(Red)-labeled microtubules advanced forward was selected from each cell. The speed of microtubule movements was calculated from the initial and final centroid positions of the photoconverted microtubule patches. The NM distance was calculated as the distance between the front of the nucleus and the centroid of the photoconverted microtubules.

### Statistical tests

All statistical analysis was performed using Prism 10 (GraphPad). Figures and figure legends contain the statistical test used and specific P values for each quantification. Kolmogorov–Smirnov test or Shapiro–Wilk test was used to determine the normality of data. Parametric tests were used for normally distributed data. F test of equality of variances was performed to determine whether an unpaired $t$ test or unpaired $t$ test with Welch's correction was used. For data that did not pass the normality test but contained a large sample number, both $t$ test and non-parametric Mann–Whitney test were performed to confirm that the same conclusions of significance were obtained.

### Online supplemental material

Fig. S1 shows the isoform analysis, motif alignment, and protein structure prediction of Nesprin-2. Fig. S2 shows the validation of Nesprin-2 mutant mouse line. Fig. S3 shows the analysis of microtubule orientation and the transport activities by Nesprin-2 SR48 mutants in peroxisome trafficking assay. Fig. S4 shows the size effects on peroxisome transport. Fig. S5 shows the relative distribution of peroxisomes with centrosome and the colocalization of mEos-DCX and microtubules in CGCs. Video 1 shows nuclear translocation in lobe culture and reaggregate culture of CGCs. Video 2 shows the predicted protein structures of SR52-KASH and SR48-KASH. Video 3 shows the impaired nuclear translocation in Nesprin-2 KO CGCs and the rescue of nuclear movements by Nesprin-2 SR48-KASH expression. Videos 4 and 5 show peroxisome trafficking assays in COS7 using Kif5B, BicD2, Nesprin-2 SR48ΔKASH, LEAA and 5A-AAVV mutants of SR48ΔKASH. Video 6 shows the movements of single peroxisomes along microtubules in MRC5. Video 7 shows the movements of peroxisomes with increased size. Videos 8 and 9 show peroxisome trafficking assays in CGCs. Video 10 shows the MT photoconversion experiment in CGCs.

### Data availability

The data are available from the corresponding author upon reasonable request.

## Acknowledgments

We thank Lucas Kapitain for plasmids and advice, Ayano Kawaguchi for guidance on i-GONAD method, Hiroki Umeshima for lobe culture method and plasmids, Toshiyuki Watanabe for live-imaging method, Kazuto Fujishima and Naotaka Nakazawa for technical assistance and plasmids, Sawako Yamashiro and Naoki Watanabe for discussion, Panpan Zhang for FACS experiment and discussion, Jaeyeon Suh for cDNA library, and Carlos Mario Rodríguez Reza for photoconversion method.

This work was supported by the Japan Society for the Promotion of Science (JSPS) Kakenhi (20H00483 and 22H05169) and Takeda Science Foundation to M. Kengaku, the JST SPRING program and Grant-in-Aid for JSPS Fellows (23KJ1280) to C. Zhou. Open Access funding provided by Kyoto University.

Author contributions: C. Zhou: Conceptualization, Data curation, Formal analysis, Funding acquisition, Investigation, Methodology, Resources, Software, Validation, Visualization, Writing—original draft, Writing—review & editing, Y.K. Wu: Investigation, Methodology, Resources, F. Ishidate: Investigation, T.K. Fujiwara: Investigation, M. Kengaku: Conceptualization, Funding acquisition, Methodology, Project administration, Resources, Supervision, Writing—original draft, Writing—review & editing.

Disclosures: The authors declare no competing interests exist.

Submitted: 6 May 2024

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

# Supplemental material

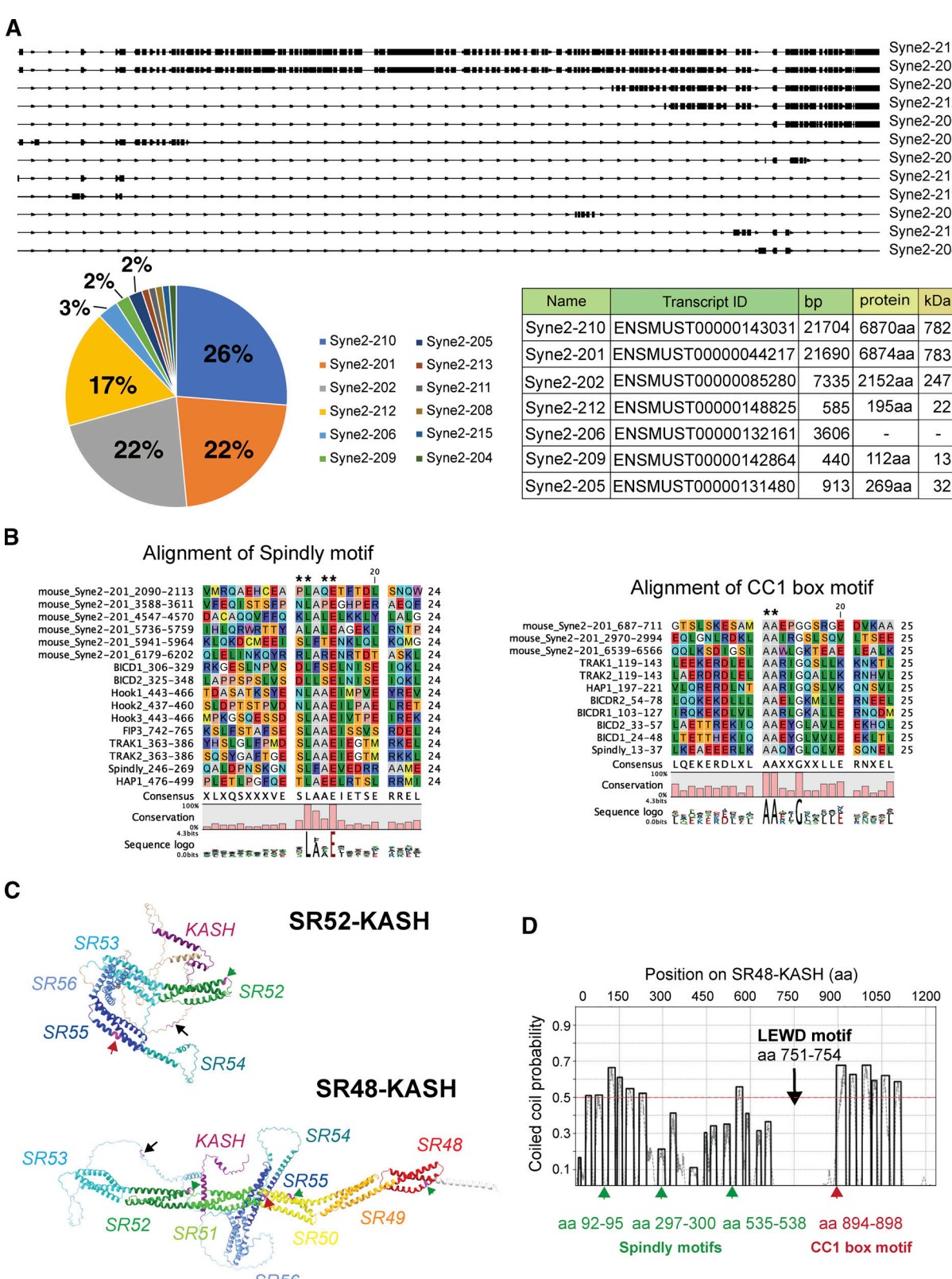

Figure S1. **Predicted isoforms and motor-binding domains of Nesprin-2. (A)** The relative abundance of Nesprin-2 transcripts was analyzed by MISO. Both Syne2-210 and Syne2-201 are translated as the full-length Nesprin-2G. Syne2-206 undergoes nonsense-mediated decay. **(B)** Sequence alignment of the putative Spindly motifs (left) and CC1 box motifs (right) of Nesprin-2 and other dynein adaptors. Asterisks indicate conserved amino acids. Adapted from Olenick and Holzbaur (2019). **(C)** Predicted protein structure of SR52-KASH and SR48-KASH by ColabFold. Different colors correspond to different SRs. Black arrows indicate the LEWD motif, green arrows indicate the putative Spindly motifs, and red arrows indicate the putative CC1 motifs. Related to Video 2. **(D)** The putative coiled-coil domains of SR48-KASH were predicted by Deepcoil2.

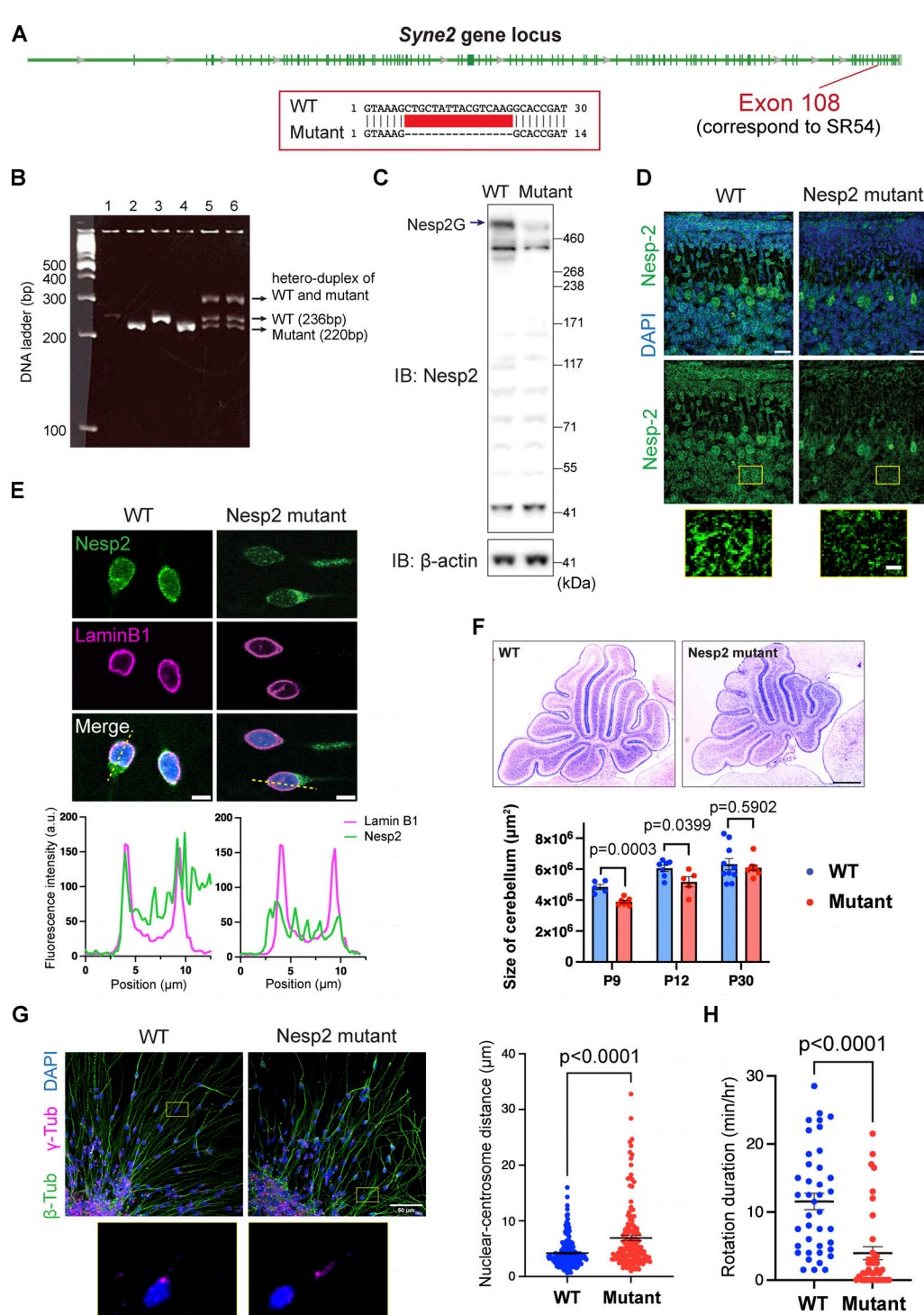

Figure S2. **Establishment and validation of a Nesprin-2 mutant mouse line. (A)** Annotation of Syne2 gene (NM_001005510.2) from NCBI RefSeq. The deletion site in the mutant mouse line is indicated. Exons are shown as green bars. **(B)** An example of genotyping results. PCR products from WT and mutant were distinguished by agarose gel electrophoresis. **(C)** Western blot of the lysates from P9 WT and mutant cerebella. **(D)** Immunofluorescence of Nesprin-2 in the cerebella from P9 WT and Nesprin2 mutant mice. The zoom-in views of the selected regions are shown. **(E)** Immunocytochemistry of CGC reaggregate cultures with Nesprin-2 (green) and Lamin B1 (magenta). The fluorescence intensity was analyzed along the dashed yellow lines. **(F)** Top: Nissl staining of mid-sagittal cryosections of P9 cerebella. Bottom: Area quantification of midsagittal sections of P9, P12, and P30 samples. $n$ = 5, 7, and 10 mice (WT), $n$ = 7, 5, and 9 mice (mutant) from three to four independent experiments. Unpaired $t$ test. **(G)** Left: Immunocytochemistry of CGC reaggregate cultures stained with β-tubulin (green) and γ-tubulin (magenta). The zoom-in views of the selected regions are shown. Right: Quantification of the nuclear-centrosome distance. $n$ = 205 (WT) and 148 (mutant) cells. Unpaired $t$ test with Welch's correction. **(H)** Quantification of nuclear rotation in CGC reaggregate cultures. During time-lapse imaging, duration of the nuclear rotation over 10° was counted. $n$ = 40 (WT) and 38 (mutant) cells from four independent experiments. Unpaired $t$ test. Bars show mean ± SEM. Scale bars, 20 μm in D upper panels; 5 μm in D lower panels, E and G lower panels; 500 μm in F; 50 μm in G upper panels. Source data are available for this figure: SourceData FS2.

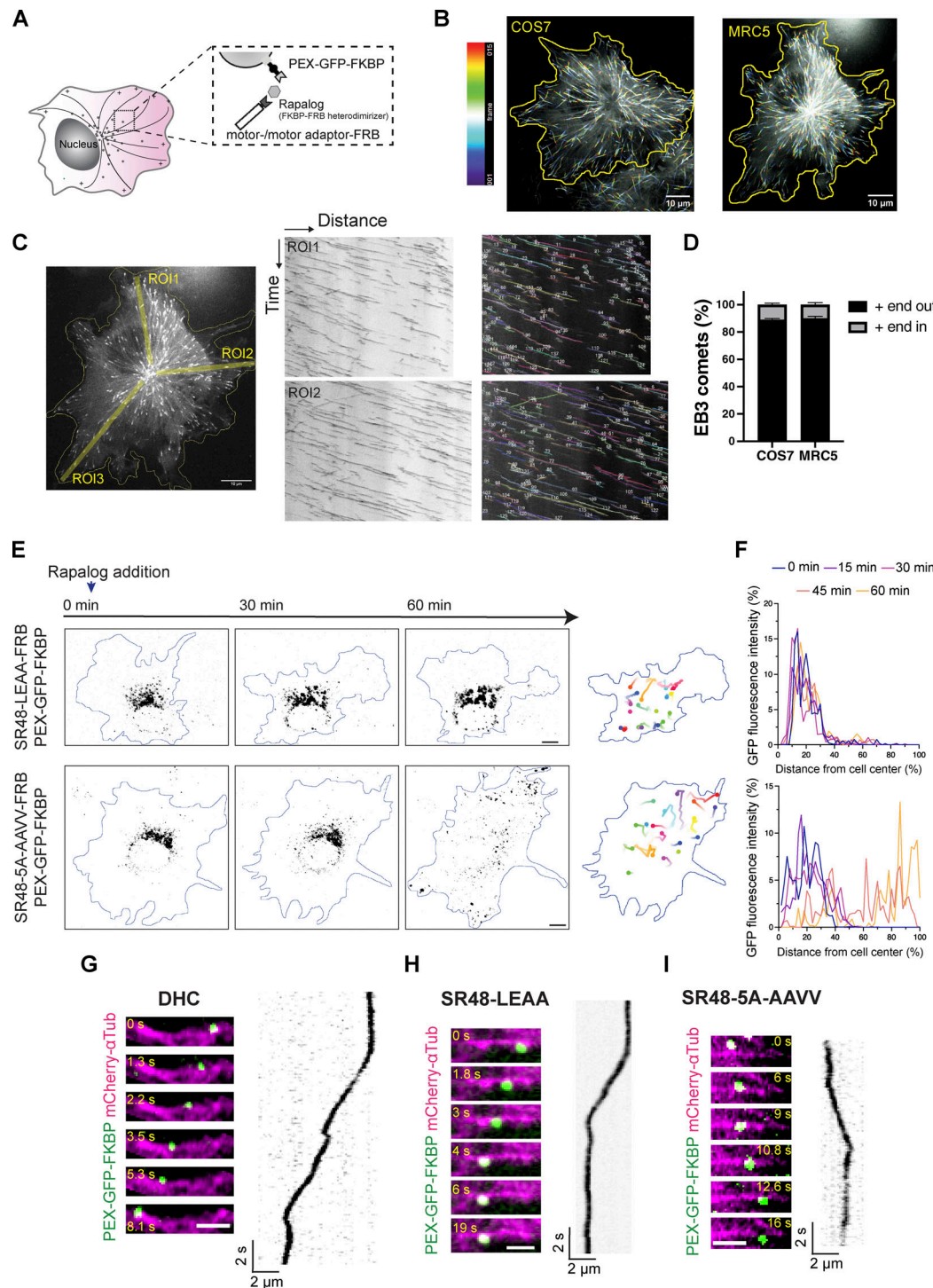

Figure S3. **Peroxisome motility was driven by Nesp2 mutants. (A)** Schematic of rapalog-inducible peroxisome trafficking assay. **(B)** Temporal color-coded time-lapse images of GFP-EB3-expressing COS7 and MRC5 cells. Images were taken at 1-s intervals. Projection of 15 frames is shown. **(C)** Analysis of EB3 comet movements. ROIs were selected for generating kymographs (right). The directions of EB3 kymographs were analyzed using KymoButler. **(D)** In both COS7 and MRC5 cells, ~90% of EB3 comets were moving toward the cell periphery. n = 15 (COS7) and 11 (MRC5) cells from three independent experiments, and over 100 trajectories were analyzed per cell. **(E)** Rapalog-induced peroxisome transport by Nesp2-SR48ΔKASH-LEAA (SR48-LEAA) (top) and Nesp2-SR48ΔKASH-5A-AAVV (SR48-5A-AAVV) (bottom). Images at 0, 30, and 60 min after rapalog treatment are shown. GFP signals are shown in black and cell contours are outlined in blue. Right: Representative trajectories of GFP signals are shown with lines by different colors, with time denoted by color gradient and ending points denoted by filled circles. **(F)** Distribution of GFP fluorescence along the distance from the cell center to the periphery at different time points. Each graph corresponds to each cell shown in E. SR48-LEAA drove limited movement of peroxisomes. SR48-5A-AAVV induced slow movement toward the cell periphery. **(G–I)** Representative time-lapse sequences (left) and kymographs (right) of rapalog-induced peroxisome transport by SR48-LEAA, SR48-5A-AAVV and DHC. Images were taken at 0.1 s interval in MRC5 cells with microtubules oriented from minus (left) to plus (right) ends. Bars show mean ± SEM. Scale bars, 10 μm in B, C, and E; 2 μm in G–I. Related to Video 5.

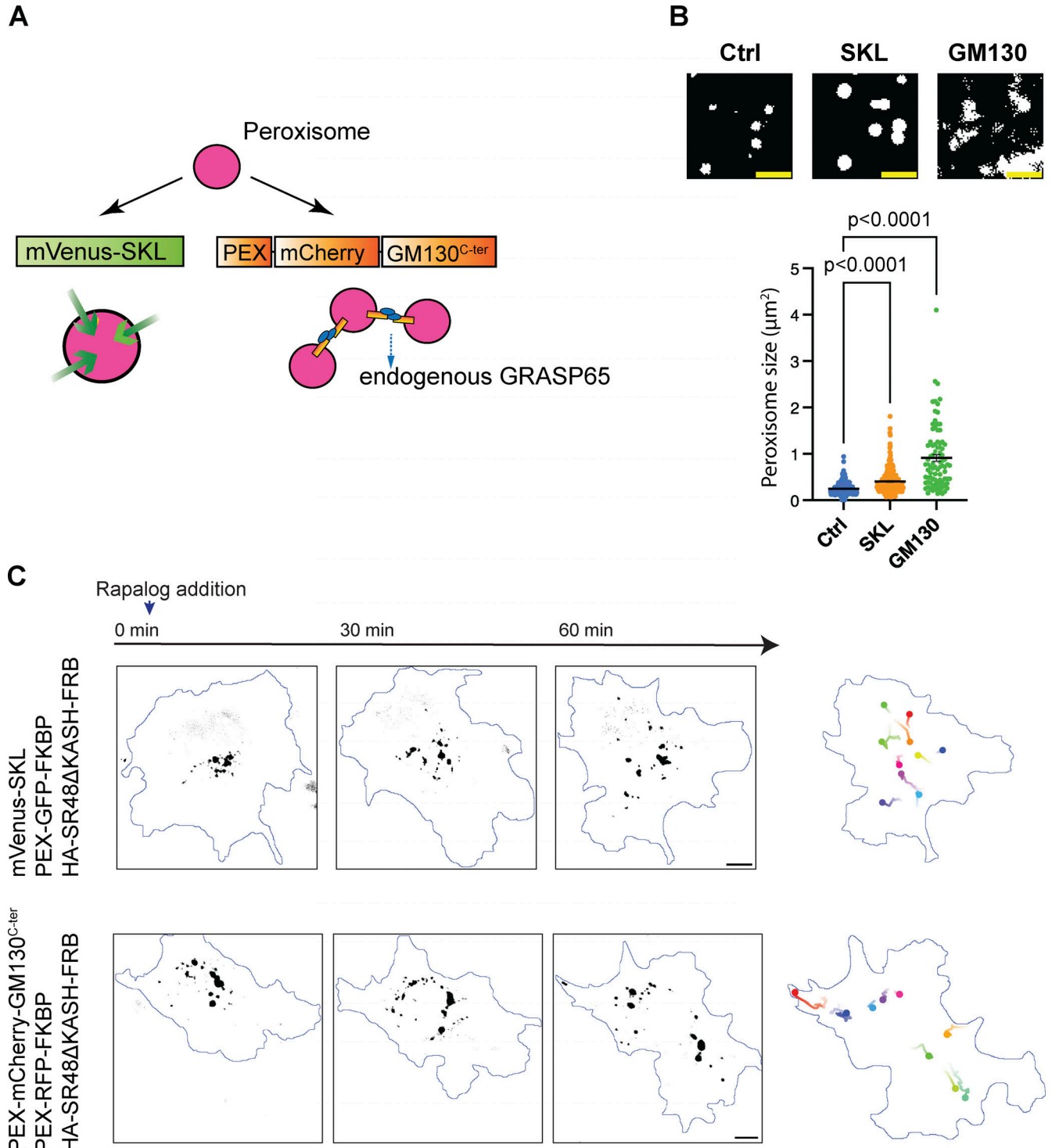

Figure S4. **Size increase of peroxisomes does not promote dynein-dominating movements by Nesprin-2. (A)** Strategies for increasing peroxisome size. SKL-tagged proteins are imported into the peroxisomes and increase the peroxisome size. GM130$^{C-ter}$ crosslinks multiple peroxisomes through homo-oligomerization of endogenous GRASP65. **(B)** Quantification of the peroxisome size. COS7 cells were transfected with PEX-GFP-FKBP only (Ctrl), PEX-GFP-FKBP and mVenus-SKL (SKL), or PEX-RFP-FKBP and PEX-mCherry-GM130$^{C-ter}$ (GM130). The area of single peroxisomes was measured. $n$ = 284 (Ctrl), 287 (SKL), and 87 (GM130) peroxisomes from four to six cells per group. Unpaired $t$ test with Welch's correction. **(C)** Rapalog-induced peroxisome transport by Nesp2-SR48ΔKASH in COS7 cells expressing mVenus-SKL or PEX-mCherry-GM130$^{C-ter}$. Representative trajectories are shown with lines of different colors. Bars show mean ± SEM. Scale bars, 2 μm in B; 10 μm in C. Related to Video 7.

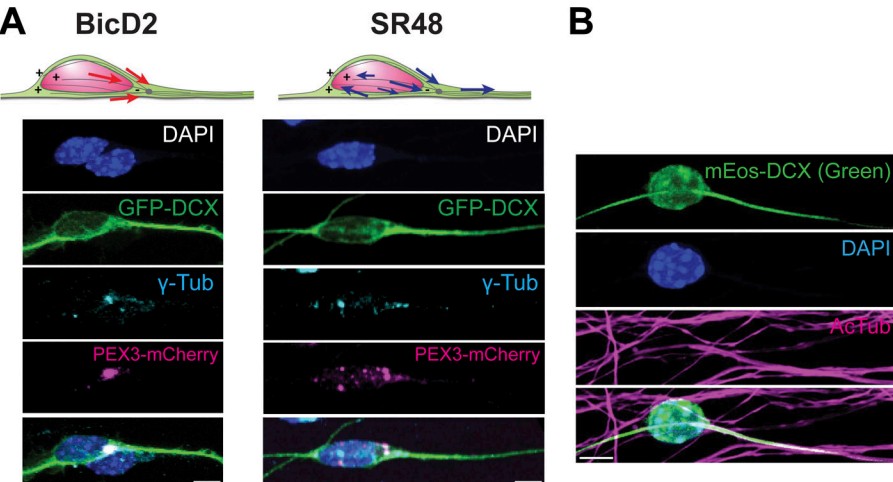

Figure S5. **γ-Tubulin staining of peroxisome assay and validation of mEos-DCX expression in CGCs. (A)** Top: Schematic of peroxisome movement driven by BicD2 or Nesp2-SR48. Bottom: immunocytochemistry with γ-tubulin (cyan) shows that BicD2 gathers peroxisomes at the centrosome, while SR48 distributes peroxisomes throughout the cell soma. **(B)** Immunocytochemistry of CGCs transfected with mEos-DCX. mEos-DCX (green) is colocalized with acetylated-tubulin (magenta). Scale bars, 5 μm.

Video 1. **Nuclear movements in lobe culture and reaggregate culture.** Time-lapse imaging was performed at 30-s intervals for 1 h. The movie is played at 20 frames per second. Scale bar, 5 μm.

Video 2. **3D view of the predicted structures of SR52-KASH and SR48-KASH.** The kinesin-binding and the putative dynein-binding sites are labeled in magenta.

Video 3. **Impaired nuclear movements in Nesprin-2 mutant CGCs was rescued by SR48KASH.** Time-lapse imaging was performed at 30-s intervals for 1 h. The movie is played at 20 frames per second. Scale bar, 5 μm.

Video 4. **Peroxisome trafficking assay of Kif5B, BicD2, and SR48 in COS7 cells.** Time-lapse imaging was performed at 30-s intervals for 1 h. Rapalog was added at 0 min. The movie is played at 20 frames per second. Scale bar, 10 μm.

Video 5. **Peroxisome trafficking assay of SR48-LEAA and SR48-5A-AAVV PEX in COS7 cells.** Time-lapse imaging was performed at 30-s intervals for 1 h. Rapalog was added at 0 min. The movie is played at 20 frames per second. Scale bar, 10 μm.

Video 6. **Dual-color imaging of peroxisome movements on MTs in MRC5 cells.** Time-lapse imaging was performed at 0.1 s intervals. The movie is played at 60 frames per second. Scale bar, 2 μm.

Video 7. **Peroxisome trafficking assay in the cell with increased cargo size.** Time-lapse imaging was performed at 30-s intervals for 1 h. Rapalog was added at 0 min. The movie is played at 20 frames per second. Scale bar, 10 μm.

Video 8.   **Peroxisome trafficking assay in CGCs.** Time-lapse imaging was performed at 30-s intervals for 1 h. Rapalog was added at 0 min. The movie is played at 20 frames per second. Scale bar, 5 µm.

Video 9.   **Dual-color imaging of MT-based PEX movements in CGCs.** Time-lapse imaging was performed at 5-s intervals for Kif5B and SR48 samples, or at 4-s intervals for the BicD2 sample. The movie is played at 60 frames per second. Scale bar, 5 µm.

Video 10.   **Nuclear translocation is accompanied by forward MT movements in migrating CGCs.** Time-lapse imaging was performed at 1-min intervals. The movie is played at 5 frames per second. Scale bar, 5 µm.

