## [Peer Review File · The Journal of Cell Biology]

Nesprin-2 coordinates opposing microtubule motors during nuclear migration in neurons

Chuying Zhou, You Wu, Fumiyoshi Ishidate, Takahiro Fujiwara, and Mineko Kengaku

Corresponding Author(s): Mineko Kengaku, Kyoto University

Review Timeline:

Submission Date:	2024-05-06
Editorial Decision:	2024-06-24
Revision Received:	2024-07-03
Editorial Decision:	2024-07-18
Revision Received:	2024-07-22

Monitoring Editor: Cassandra Ori-McKenney

Scientific Editor: Tim Fessenden

Transaction Report:

DOI: <https://doi.org/10.1083/jcb.202405032>

June 24, 2024

Re: JCB manuscript #202405032

Prof. Mineko Kengaku
Kyoto University
Institute for Integrated Cell-Material Sciences
Yoshida Honmachi
Sakyo-ku
Kyoto 606-8501
Japan

Dear Prof. Kengaku,

Thank you for submitting your manuscript entitled "Nesprin-2 coordinates opposing microtubule motors during nuclear migration in neurons". The manuscript was assessed by expert reviewers, whose comments are appended to this letter. We appreciate your patience during the unusually long review process. We invite you to submit a revision if you can address the reviewers' key concerns, as outlined here.

As you will see, both reviewers appreciated the conceptual advance made in this work on the interactions of both kinesin and dynein motors with Nesprin-2 and the ramifications for nuclear movements *in vivo*. Reviewer 1 raised concerns that the work leaves the requirement for the activating adaptor BicD2 somewhat unclear, and feels the final model for how both types of motors bind to Nesprin-2 may be inconsistent with data provided. To the same point, this reviewer also sought clarification on the proposed coiled-coil domains in Nesprin-2. A suitably revised manuscript should better align the conclusions with observations made on this question.

GENERAL GUIDELINES:

Text limits: Character count for an Article is < 40,000, not including spaces. Count includes title page, abstract, introduction, results, discussion, and acknowledgments. Count does not include materials and methods, figure legends, references, tables, or supplemental legends.

Figures: Articles may have up to 10 main text figures. Figures must be prepared according to the policies outlined in our Instructions to Authors, under Data Presentation, <https://jcb.rupress.org/site/misc/ifora.xhtml>. All figures in accepted manuscripts will be screened prior to publication.

*****IMPORTANT:** It is JCB policy that if requested, original data images must be made available. Failure to provide original images upon request will result in unavoidable delays in publication. Please ensure that you have access to all original microscopy and blot data images before submitting your revision. ***

Supplemental information: There are strict limits on the allowable amount of supplemental data. Articles may have up to 5 supplemental figures. Up to 10 supplemental videos or flash animations are allowed. A summary of all supplemental material should appear at the end of the Materials and methods section.

Please note that JCB now requires authors to submit Source Data used to generate figures containing gels and Western blots with all revised manuscripts. This Source Data consists of fully uncropped and unprocessed images for each gel/blot displayed in the main and supplemental figures. Since your paper includes cropped gel and/or blot images, please be sure to provide one Source Data file for each figure that contains gels and/or blots along with your revised manuscript files. File names for Source Data figures should be alphanumeric without any spaces or special characters (i.e., SourceDataF#, where F# refers to the associated main figure number or SourceDataFS# for those associated with Supplementary figures). The lanes of the gels/blots should be labeled as they are in the associated figure, the place where cropping was applied should be marked (with a box), and molecular weight/size standards should be labeled wherever possible.

The typical timeframe for revisions is three to four months. While most universities and institutes have reopened labs and allowed researchers to begin working at nearly pre-pandemic levels, we at JCB realize that the lingering effects of the COVID-19 pandemic may still be impacting some aspects of your work, including the acquisition of equipment and reagents. Therefore, if you anticipate any difficulties in meeting this aforementioned revision time limit, please contact us and we can work with you to find an appropriate time frame for resubmission. Please note that papers are generally considered through only one revision cycle, so any revised manuscript will likely be either accepted or rejected.

Thank you for this interesting contribution to Journal of Cell Biology. You can contact us at the journal office with any questions at cellbio@rockefeller.edu.

Sincerely,

Kassandra Ori-McKenney
Monitoring Editor
Journal of Cell Biology

Tim Fessenden
Scientific Editor
Journal of Cell Biology

Reviewer #1 (Comments to the Authors (Required)):

The paper by Zhou et al. reports a molecular analysis of the nuclear envelope protein nesprin-2 in its role as a coordinator of microtubule motor protein function during nuclear migration in developing neurons. While this role for nesprin-2 has been reported by previous research, including from this lab, Zhou et al. perform more in-depth molecular analysis of domains and sequences necessary for molecular motor interaction and function. They conclude that nesprin-2 integrates both dynein and kinesin-1 functions during nuclear migration, in support of previously published data and models. They demonstrate the specific regions of the protein necessary for these functions, also largely in support of prior work. Their conclusions about integration of dynein and kinesin motor activities differ from some previous reports that suggested dynein and kinesin play antagonistic roles during nuclear migration in developing neurons. The domain analysis is very thorough but does not provide strikingly new information than previous reports. The experiments appear to be of high quality and the imaging is very nice and convincing. I do not have strong technical concerns about the experiments or data analysis. I think the paper would be of interest to the cell biology, cytoskeleton, nuclear biology, and neuronal biology fields. I am supportive of publication of a suitably revised manuscript.

My biggest concern with the paper is the interpretation of data surrounding the role of nesprin as a direct or indirect adapter for the dynein motor. Previous work, and the current study, has shown consistently that nesprin-2 interacts with the well-characterized dynein adapter protein BicD2. Through this interaction, it is logical to assume that nesprin-2 is capable of recruiting activated dynein-dynactin-BicD2 (DDB) complexes to the nuclear envelope, and indeed the authors conclude the same in the current study. However, through sequence analysis, they also suggest that nesprin-2 contains its own dynein and dynactin interaction motifs, the spindly and CC1 box sequences. Of note, these sequences are of low sequence complexity and there may be a fairly high probability of finding similar sequences in long stretches of amino acids. The authors then seem to suggest a model that nesprin-2 may be directly scaffolding dynein and dynactin as an activating adapter protein. Yet their own data show that BicD2 binding to nesprin-2 is required for dynein recruitment and activity. This is confusing. It seems to me that the authors own data, combined with previously published data, strongly suggest that BicD2 is the dynein activating adapter that recruits dynein and dynactin to nesprin-2. It is therefore unclear why the authors suggest the presence of independent dynein-dynactin binding sequences within nesprin-2, despite the fact that their own data suggests these sequences are not sufficient to recruit dynein-dynactin in the absence of BicD2. I suggest the authors reconsider their analysis of spindly and CC1 boxes within the nesprin-2 sequence and re-frame their results to more accurately reflect their own data that nesprin-2's interaction with BicD2 is what underlies its ability to recruit active dynein-dynactin. Further, I do not find the argument that this region of nesprin-2 contains a coiled-coil sequence that might act in a similar manner as other dynein adapter proteins credible based on the authors own data in Fig. S1 (see below). These sequences and the structural model look like alpha-helical spectrin repeats to me.

For instance, the authors conclude that "Although the detailed mechanism remains elusive, our results indicate that dynein-

dynein binding to Nesprin-2 is dependent on the Spindly and CC1 box motifs." However, their own data show that in fact the dynein-dynactin binding to nesprin-2 is dependent on BicD2 (Fig. 2), leading to confusing messaging on the model they are proposing. The authors should reconsider how strong their data is supporting a model whereby nesprin-2 binds directly to dynein-dynactin. If they would like to make this conclusion, please supply new data that shows BicD2-independent dynein-dynactin interactions with nesprin-2.

Minor points:

1. Lines 402-403: Please expand on this comment. I am not aware that JIP1 or TRAK2 contain LEWD motifs.
2. The paper is long and the authors may consider reducing the number of main figures. For instance, Figure 1 recapitulates most of what has been published in their prior paper, Wu et al. 2018, and the authors may consider moving it to the supplement. Couldn't Figure 3 and 4 be combined?
3. In Fig. S1A and the manuscript, the authors argue that there is a potential coiled-coil in the vicinity of the spindly and CC1 motifs located in proximity to the LEWD motif. However, the AF2 structure predictions do not support this conclusion, as no coiled-coil is observed. Rather, folded spectrin repeats are apparent in these structures. So it is unclear to me why the authors still conclude that a coiled-coil is present in this region of the protein. Indeed, the coiled-coil prediction algorithm finds only a moderate chance of coiled-coil probability in this region (Fig. S1D). The authors should expand on how they conclude that a coiled-coil structure is likely in this region of the nesprin protein.
4. The data from nesprin-2 KO mouse in Fig. 3 doesn't appear strongly connected to the main story and there should probably be more context on how to interpret these findings for readers who are not specialists in brain development. The effects are relatively mild and I wonder if this figure could also be moved to the supplement to streamline the main story.

Reviewer #2 (Comments to the Authors (Required)):

In this manuscript, the authors show that nesprin-2 can simultaneously interact with both kinesin and dynein, and that nesprin-2 leads to cooperation of the motors to move nuclei in migrating neurons. The implications of these experiments are broad, leading to a better cellular and mechanistic understanding of nuclear movements in neurons, defects in which lead to a variety of disease. The strength of the manuscript is that it combines live imaging in cerebellar cultured sections or CGCs, in vitro GST pull down assays, an in vivo phenotypic description of a new mouse knockout, and a heterologous COS7 melanosome movement system. This powerful combination of results leads to a mechanistic understanding of how motors cooperate to move nuclei.

This manuscript is clearly written, and the experiments are all well performed with appropriate controls. The combination of experimental approaches is strong. I particularly appreciated the new mutant mouse generated here and the rescue experiments in the CGCs. This manuscript will be of interest to a broad group of cell biologists and neuro developmental biologists.

I only have a handful of very minor comments:

I wouldn't call the new mouse model a nesprin-2 knockout. Nesprin-2 has so many different isoforms, it's nearly impossible to say a mutation in the SR54 leads to knock out of the entire nesprin-2 gene. Calling previous mouse models that deleted just the KASH domain as knockouts has set the field back. Be specific here, it is a mutation that disrupts the part of nesprin-2 that interacts with microtubule motors. Many other nesprin-2 isoforms are likely still present.

Lines 158-166 call out to Fig S2, but I think they mean Fig S1.

I also think the line 158-9 is a bit of an overinterpretation of the Colabfold result. Spectrin proteins are rarely extended in Colabfold.

Lines 317-318. While I agree with this conclusion, it feels a little too strongly worded to me.

Reviewer #1

Thank you for giving us the opportunity to submit a revised manuscript. We really appreciate your thorough and thoughtful consideration of our manuscript. Your review has provided many helpful suggestions, which have greatly improved our paper. In response to these comments, as well as those of the other Reviewers, we have re-considered the conclusion from some of our results and have rewritten the manuscript. We hope that these as well as our point-by-point responses below will answer any uncertainties and be to the satisfaction of the Reviewers.

Major points: ... *I suggest the authors reconsider their analysis of spindly and CC1 boxes within the nesprin-2 sequence and re-frame their results to more accurately reflect their own data that nesprin-2's interaction with BicD2 is what underlies its ability to recruit active dynein-dynactin... The authors should reconsider how strong their data is supporting a model whereby nesprin-2 binds directly to dynein-dynactin. If they would like to make this conclusion, please supply new data that shows BicD2 independent dynein-dynactin interactions with nesprin-2.*

Response: We thank the reviewer for pointing out this issue. After careful re-evaluation of the predicted structure of Nesprin-2 and the co-immunoprecipitation results from Fig. 2 G-K, we agree that our data support the interaction between Nesprin-2 and dynein-dynactin complex is more likely mediated by BicD2 binding. Although the Spindly and CC1 box motifs which we found to be critical for the binding of dynein-dynactin complex have not been identified as BicD2 binding sites, our data using BicD2-deleted cells clearly indicated that the interaction of Nesprin-2 to dynein-dynactin, but not kinesin, requires BicD2. We have revised the relevant parts of Results and Discussion (lines 192-195, 390-399).

Minor points:

1. Lines 402-403: Please expand on this comment. I am not aware that JIP1 or TRAK2 contain LEWD motifs.

Response: We thank the reviewer for pointing that the original writing was misleading. We intended to discuss that the dynein-interacting domains are physically close to kinesin-interacting domains, enabling interplay between bidirectional motors. Indeed, JIP1 recruits KLC through its C-terminal (PTEDIYLE) region, a different mechanism from LEWD-motif-mediated KLC recruitment (Verhey et al., 2001). TRAK2 recruits KIF5B through

its N-terminus via unknown motif (van Spronsen et al., 2013; Canty et al., 2023). We have rephrased the corresponding sentence and highlighted in the main text (lines 403-405).

2. The paper is long and the authors may consider reducing the number of main figures. For instance, Figure 1 recapitulates most of what has been published in their prior paper, Wu et al. 2018, and the authors may consider moving it to the supplement. Couldn't Figure 3 and 4 be combined?

Response: Thank you for the suggestion. We have combined Figure 3 and 4 by replacing some data (the original Figure 3A and 3B) to the Supplementary Fig. S2. We have also deleted the original Figure 1G, 1H, 1M and 1N to reduce the panels. Since we have already used the upper limit of 5 supplementary figures, we have decided to retain the other panels as Figure 1. Although the new Figure 1J is recapitulation of our previous manuscript (Fig. 5C of Wu et al., 2018), the kinetic analysis in the tissue in the panels 1A-1F and the observation of the frequent backward steps in 1E and 1I are new data, which we believe would lead to the analysis of bidirectional motors from Figure 2 onward. We have revised the text to clarify new data and reproduced data.

3. In Fig. S1A and the manuscript, the authors argue that there is a potential coiled-coil in the vicinity of the spindly and CC1 box motifs located in proximity to the LEWD motif. However, the AF2 structure predictions do not support this conclusion, as no coiled-coil is observed. Rather, folded spectrin repeats are apparent in these structures. So it is unclear to me why the authors still conclude that a coiled-coil is present in this region of the protein. Indeed, the coiled-coil prediction algorithm finds only a moderate chance of coiled-coil probability in this region (Fig. S1D). The authors should expand on how they conclude that a coiled-coil structure is likely in this region of the nesprin protein.

Response: We appreciate the important point raised by the reviewer. We agree with the reviewer that the description inferring the molecular mechanism of dynein-dynactin binding in comparison with known structures was inadvertent, and the predicted 33-nm stretch in the SR48-53 region is unlikely to contain coiled-coils. We have reconsidered the results and have rewritten the interpretation of the structure prediction (lines 159-170). The correction is highlighted in the main text.

4. The data from nesprin-2 KO mouse in Fig. 3 doesn't appear strongly connected to the

main story and there should probably be more context on how to interpret these findings for readers who are not specialists in brain development. The effects are relatively mild and I wonder if this figure could also be moved to the supplement to streamline the main story.

Response: We have moved the original Fig. 3A and 3B to supplemental Fig. S2 and combined Figs. 3 and 4 to make the main story more compact.

Reviewer #2

We would like to thank the Reviewer for their thorough and thoughtful consideration of our manuscript. Your review has provided many helpful suggestions, which have greatly improved our paper. In response to these comments, as well as those of the other Reviewers, we have re-considered the conclusion from some of our results and have rewritten the manuscript. We hope that these as well as our point-by-point responses below will answer any uncertainties and be to the satisfaction of the Reviewers.

Comment 1: *I wouldn't call the new mouse model a nesprin-2 knockout. Nesprin-2 has so many different isoforms, it's nearly impossible to say a mutation in the SR54 leads to knock out of the entire nesprin-2 gene. Calling previous mouse models that deleted just the KASH domain as knockouts has set the field back. Be specific here, it is a mutation that disrupts the part of nesprin-2 that interacts with microtubule motors. Many other nesprin-2 isoforms are likely still present.*

Response: Thank you for pointing out the inaccurate description. Indeed, we anticipated that our mutation model generates a truncated form of Nesprin-2 without the C-terminal KASH domain and disrupts the localization of Nesprin-2 onto the nuclear envelope, which is supported by immunostaining result in Fig. S2 E. Moreover, the overall expression level of full-length Nesprin-2 also decreased (Fig. S2, C-E), which might be due to nonsense-mediated decay of Nesprin-2 mRNA. We agree that many other Nesprin-2 isoforms lacking the C-terminal transmembrane region are probably still present. We have rephrased the way we address the nesprin-2 mutant model in the main text and highlighted the correction.

Comment 2: *Lines 158-166 call out to Fig S2, but I think they mean Fig S1.*

Response: Thank you pointing out the typo. We have corrected accordingly (lines 162-168).

Comment 3: *I also think the line 158-9 is a bit of an overinterpretation of the Colabfold result. Spectrin proteins are rarely extended in Colabfold.*

Response: We agree with the reviewer that the predicted 33-nm stretch in the SR48-53 region is unlikely to contain coiled-coils. We have revised the interpretation of ColabFold results, which is highlighted in the main text (lines 159-170).

Comment 4: *Lines 317-318. While I agree with this conclusion, it feels a little too strongly worded to me.*

Response: We thank the reviewer for pointing this out. We have made the change, and the new sentence reads as follow:

These results did not support the cooperative interplay of motors bound to multiple Nesprin molecules. (lines 318-319)

All of the minor points the reviewer had raised have been accommodated in the revised manuscript.

July 18, 2024

RE: JCB Manuscript #202405032R

Prof. Mineko Kengaku
Kyoto University
Institute for Integrated Cell-Material Sciences
Yoshida Honmachi
Sakyo-ku
Kyoto 606-8501
Japan

Dear Prof. Kengaku:

Thank you for submitting your revised manuscript entitled "Nesprin-2 coordinates opposing microtubule motors during nuclear migration in neurons". As you will see, Reviewer 1 recommends this work for publication with changes to the text concerning the sequences identified in Nesprin-2. We agree that discussion of these sequences should endeavor to avoid confusion, especially in relation to prior literature, and that doing so does not diminish the novelty and significance of these findings. The arrangement of figures commented on by this reviewer is left to your discretion. We would be happy to publish your paper in JCB pending these final changes and revisions necessary to meet our formatting guidelines (see details below).

A. MANUSCRIPT ORGANIZATION AND FORMATTING:

Full guidelines are available on our Instructions for Authors page, <http://jcb.rupress.org/submission-guidelines#revised>. Submission of a paper that does not conform to JCB guidelines will delay the acceptance of your manuscript.

1) Text limits: Character count for Articles is < 40,000, not including spaces. Count includes abstract, introduction, results, discussion, and acknowledgments. Count does not include title page, figure legends, materials and methods, references, tables, or supplemental legends.

2) Figures limits: Articles may have up to 10 main figures and 5 supplemental figures/tables.

3) Figure formatting: Scale bars must be present on all microscopy images, including inset magnifications. Molecular weight or nucleic acid size markers must be included on all gel electrophoresis. Please avoid pairing red and green for images and graphs to ensure legibility for color-blind readers. If red and green are paired for images, please ensure that the particular red and green hues used in micrographs are distinctive with any of the colorblind types. If not, please modify colors accordingly or provide separate images of the individual channels.

4) Statistical analysis: Error bars on graphic representations of numerical data must be clearly described in the figure legend. The number of independent data points (n) represented in a graph must be indicated in the legend. Statistical methods should be explained in full in the materials and methods. For figures presenting pooled data the statistical measure should be defined in the figure legends. Please also be sure to indicate the statistical tests used in each of your experiments (either in the figure legend itself or in a separate methods section) as well as the parameters of the test (for example, if you ran a t-test, please indicate if it was one- or two-sided, etc.). Also, if you used parametric tests, please indicate if the data distribution was tested for normality (and if so, how). If not, you must state something to the effect that "Data distribution was assumed to be normal but this was not formally tested."

** Please indicate n for independent data points shown in Figures S2 and S3.

5) Abstract and title: The abstract should be no longer than 160 words and should communicate the significance of the paper for a general audience. The title should be less than 100 characters including spaces. Make the title concise but accessible to a general readership.

6) Materials and methods: Should be comprehensive and not simply reference a previous publication for details on how an experiment was performed. Please provide full descriptions in the text for readers who may not have access to referenced manuscripts. We also provide a report from SciScore and an associate score, which we encourage you to use as a means of evaluating and improving the methods section.

7) Please be sure to provide the sequences for all of your primers/oligos and RNAi constructs in the materials and methods. You must also indicate in the methods the source, species, and catalog numbers (where appropriate) for all of your antibodies.

Please also indicate the acquisition and quantification methods for immunoblotting/western blots.

8) Microscope image acquisition: The following information must be provided about the acquisition and processing of images:

- a. Make and model of microscope
- b. Type, magnification, and numerical aperture of the objective lenses
- c. Temperature
- d. Imaging medium
- e. Fluorochromes
- f. Camera make and model
- g. Acquisition software
- h. Any software used for image processing subsequent to data acquisition. Please include details and types of operations involved (e.g., type of deconvolution, 3D reconstitutions, surface or volume rendering, gamma adjustments, etc.).

10) Supplemental materials: There are strict limits on the allowable amount of supplemental data. Articles may have up to 5 supplemental figures. Please also note that tables, like figures, should be provided as individual, editable files. A summary of all supplemental material should appear at the end of the Materials and methods section.

13) ORCID IDs: ORCID IDs are unique identifiers allowing researchers to create a record of their various scholarly contributions in a single place. At resubmission of your final files, please provide an ORCID ID for all authors.

15) A data availability statement is required for all research article submissions. The statement should address all data underlying the research presented in the manuscript. Please visit the JCB instructions for authors for guidelines and examples of statements at (<https://rupress.org/jcb/pages/editorial-policies#data-availability-statement>).

Please note that JCB requires authors to submit Source Data used to generate figures containing gels and Western blots with all revised manuscripts. This Source Data consists of fully uncropped and unprocessed images for each gel/blot displayed in the main and supplemental figures. Since your paper includes cropped gel and/or blot images, please be sure to provide one Source Data file for each figure that contains gels and/or blots along with your revised manuscript files. File names for Source Data figures should be alphanumeric without any spaces or special characters (i.e., SourceDataF#, where F# refers to the associated main figure number or SourceDataFS# for those associated with Supplementary figures). The lanes of the gels/blots should be labeled as they are in the associated figure, the place where cropping was applied should be marked (with a box), and molecular weight/size standards should be labeled wherever possible. Source Data files will be directly linked to specific figures in the published article.

Journal of Cell Biology now requires a data availability statement for all research article submissions. These statements will be published in the article directly above the Acknowledgments. The statement should address all data underlying the research presented in the manuscript. Please visit the JCB instructions for authors for guidelines and examples of statements at (<https://rupress.org/jcb/pages/editorial-policies#data-availability-statement>).

B. FINAL FILES:

Thank you for your attention to these final processing requirements. Please revise and format the manuscript and upload materials within 7 days. If you need an extension for whatever reason, please let us know and we can work with you to determine a suitable revision period.

Thank you for this interesting contribution, we look forward to publishing your paper in Journal of Cell Biology.

Sincerely,

Kassandra Ori-McKenney
Monitoring Editor
Journal of Cell Biology

Tim Fessenden
Scientific Editor
Journal of Cell Biology

Reviewer #1 (Comments to the Authors (Required)):

The revised manuscript has addressed a lot of my concern but I am still worried about the author's presentation of the putative Spindly and CC1 box motifs they have identified in nesprin-2. They provide no evidence that these motifs act in a canonical manner to directly contact dynein and dynactin, but rather their evidence strongly supports a model whereby these regions are necessary for BicD2 binding. BicD2 itself contains canonical Spindly and CC1 box motifs, so it is more unclear why nesprin would have independent and redundant motifs. I think the paper should be published, but urge the authors to further refine their writing to clarify this point. Here are some specific suggestions prior to publication:

- It is unclear to me why the data in Fig. 4 and 5 are split up. The data all refer to the same experiments and should be found in the same figure in my opinion. Further, I am not sure there is a lot of deep insight in the velocity and run-length analyses in current Fig. 5, and they are better suited for a supplemental figure in my opinion.
- The authors maintain that they've identified Spindly and CC1 box motifs within nesprin-2, although the arrangement of these sequences is very different from known dynein-dynactin adapters. I am not very convinced that the sequences they've identified in nesprin-2 are acting the way that these sequences are known to act in dynein-dynactin adapter proteins. Further, the authors show clearly that BicD2 is required for dynein-dynactin binding to nesprin-2. Therefore, I still find their assignments of these sequences as Spindly and CC1 box motifs confusing, and I think the field will as well. For instance, on line 499, the authors

suggest these sequences are necessary to bind to BicD2. However, there is no known structural explanation for why Spindly and CC1 box sequences would be necessary to bind to BicD2, which itself contains these sequences. I'd urge the authors to consider naming these sequences a "putative" Spindly and CC1 box motifs throughout their work to demarcate them from verified Spindly and CC1 box motifs in known dynein-dynactin adapters. The authors data clearly shows these sequences are important, but mainly for BicD2 binding, which is not consistent with the current model for how Spindly and CC1 box motifs bind to dynein-dynactin. I think it is more likely these sequences are necessary for a structural conformation of nesprin-2 that is necessary for BicD2 binding and subsequent recruitment of dynein-dynactin.

Kengaku Laboratory
Institute for Integrated Cell-Material Sciences(iCeMS)
Kyoto University
iCeMS Research Building
Yoshida Honmachi, Sakyo-ku, Kyoto, 606-8501, JAPAN
TEL:075-753-9832 FAX:075-753-9820

July 22, 2024

Dear Dr. Fessenden,

I am sending herewith the revised version of our manuscript (#202405032R) entitled 'Nesprin-2 coordinates opposing microtubule motors during nuclear migration in neurons' for consideration for publication in *JCB*.

We appreciate the invaluable advice from Reviewer #1. To clarify that the identified motifs do not function like typical motifs, we have named them as putative motifs as suggested by the reviewer, and a few sentences have been added to Discussion. We understand the reviewer's suggestion to combine Figures 4 and 5. Still, we have decided to keep them in the original arrangement, as combining the two figures would make it too large, and the number of supplementary figures has already reached the maximum.

We sincerely hope that you will find the revised manuscript suitable for publication in *JCB*.

Sincerely yours,

Mineko KENGAKU, Ph.D.

Institute for Integrated Cell-Material Sciences
Kyoto University
Yoshida-Honmachi, Sakyo-ku
Kyoto 606-8501, Japan
Tel:+ 81-75-753-9832
Fax:+81-75-753-9820
kengaku@icems.kyoto-u.ac.jp